

# initMIP-Antarctica: An ice sheet model initialization experiment of ISMIP6

Hélène Seroussi[1], Sophie Nowicki[2], Erika Simon[2], Ayako Abe Ouchi[3], Torsten Albrecht[4], Julien Brondex[5], Stephen Cornford[6], Christophe Dumas[7], Fabien Gillet-Chaulet[5], Heiko Goelzer[8,9], Nicholas R. Golledge[10], Jonathan M. Gregory[11], Ralph Greve[12], Matthew J. Hoffman[13], Angelika Humbert[14,15], Philippe Huybrechts[16], Thomas Kleiner[14], Eric Larour[1], Gunther Leguy[17], William H. Lipscomb[17], Daniel Lowry[10], Matthias Mengel[4], Mathieu Morlighem[18], Frank Pattyn[9], Anthony J. Payne[19], David Pollard[20], Stephen Price[13], Aurélien Quiquet[7], Thomas Reerink[8], Ronja Reese[4], Christian B. Rodehacke[21,14], Nicole-Jeanne Schlegel[1], Andrew Shepherd[22], Sainan Sun[9], Johannes Sutter[14], Jonas Van Breedam[16], Roderik S.W. van de Wal[8], Ricarda Winkelmann[4], and Tong Zhang[13]

[1]Jet Propulsion Laboratory, California Institute of Technology, Pasadena, CA, USA
[2]NASA Goddard Space Flight Center, Greenbelt, MD, USA
[3]University of Tokyo, Japan
[4]Potsdam Institute for Climate Impact Research, Germany
[5]Univ. Grenoble Alpes, CNRS, IRD, Grenoble INP, IGE, 38000 Grenoble, France
[6]Swansea University, United Kingdom
[7]Laboratoire des Sciences du Climat et de l'Environnement, LSCE/IPSL, CEA-CNRS-UVSQ, Université Paris-Saclay, 91191 Gif-sur-Yvette, France
[8]Institute for Marine and Atmospheric research Utrecht, Utrecht University, The Netherlands
[9]Laboratoire de Glaciologie, Université libre de Bruxelles, Belgium
[10]Antarctic Research Centre, Victoria University of Wellington, New Zealand
[11]University of Reading, United Kingdom
[12]Institute of Low Temperature Science, Hokkaido University, Sapporo, Japan
[13]Los Alamos National Laboratory, NM, USA
[14]Alfred Wegener Institute Helmholtz Centre for Polar and Marine Research, Bremerhaven, Germany
[15]University of Bremen, Germany
[16]Earth System Science & Departement Geografie, Vrije Universiteit Brussel, Belgium
[17]National Center for Atmospheric Research, Boulder, CO, USA
[18]Department of Earth System Science, University of California Irvine, Irvine, CA, USA
[19]University of Bristol, United Kingdom
[20]Earth and Environmental Systems Institute, Pennsylvania State University, University Park, PA, USA
[21]Danish Meteorological Institute, Copenhagen, Denmark
[22]University of Leeds, United Kingdom

*Correspondence to*: Helene Seroussi (Helene.Seroussi@jpl.nasa.gov)    ©2019 - All rights reserved

**Abstract.** Ice sheet numerical modeling is the best approach to estimate the dynamic contribution of the Antarctic ice sheet to sea level rise over the coming centuries. The influence of initial conditions on ice sheet model simulations, however, is still unclear. To better understand this influence, an initial state intercomparison exercise (initMIP) has been developed to compare, evaluate, and improve initialization procedures and estimate their impact on century-scale simulations. initMIP is the first set of experiments of the Ice Sheet Model Intercomparison Project for CMIP6 (ISMIP6), which is the primary Coupled Model Intercomparison Project Phase 6 (CMIP6) activity focusing on the Greenland and Antarctic ice sheets. Following initMIP-Greenland, initMIP-Antarctica has been designed to explore uncertainties associated with model initialization and spin-up, and to evaluate the impact of changes in external forcings. Starting from the state of the Antarctic ice sheet at the end of the initialization procedure, three forward experiments are each run for 100 years: a control run, a run with a surface mass balance (SMB) anomaly, and a run with a basal melting anomaly beneath floating ice. This study presents the results of initMIP-Antarctica from 25 simulations performed by 16 international modeling groups. The submitted results use different initial conditions and initialization methods, as well as ice flow model parameters and reference external forcings. We find a good



agreement among model responses to the SMB anomaly, but large variations in responses to the basal melting anomaly. These variations can be attributed to differences in the extent of ice shelves and their upstream tributaries, the numerical treatment of grounding line, as well as the initial ocean conditions applied, suggesting that ongoing efforts to better represent ice shelves in continental-scale models should continue.

**1 Introduction**

The Antarctic ice sheet is the largest reservoir of freshwater on Earth and contains enough ice to raise global mean sea level by 58.3 m (Fretwell et al., 2013). Reconstructions of past sea-level variations show that the volume of the Antarctic ice sheet has varied significantly over time, with for example an ice loss of up to 15 m sea level equivalent (SLE) at a rate of up to 1 mm/yr during the Pliocene, around 5.3-2.6 million years before present (Miller et al., 2012). Several regions of the Antarctic ice sheet

are currently changing rapidly (Rott et al., 2002; Scambos et al., 2004; De Angelis and Skvarca, 2003; Khazendar et al., 2013; Mouginot et al., 2014; Rignot et al., 2014; Christie et al., 2016). These changes have been attributed to changes in ocean circulation (e.g., Thomas et al., 2004; Payne et al., 2004; Jenkins et al., 2010; Jacobs et al., 2012; Jenkins et al., 2018) and atmospheric conditions (e.g., Doake and Vaughan, 1991; Vaughan and Doake, 1996; Scambos et al., 2000). Understanding how the Antarctic ice sheet will evolve over the coming centuries, and in particular how much it will contribute to sea level, has

therefore become a major field of research.

Projections of 21[st] century Antarctic ice sheet evolution, however, vary widely, with projected upper bounds ranging from 30 cm of sea level equivalent (Ritz et al., 2015) to over 1 m (DeConto and Pollard, 2016), depending on model characteristics and physical processes, as well as the climate scenarios adopted. Previous efforts from the ice sheet modeling community for the IPCC-AR5 (Intergovernmental Panel for Climate Change-Fifth Assessment Report, Church et al., 2013) tried to estimate the ice

sheet evolution under several climate scenarios (Bindschadler et al., 2013; Nowicki et al., 2013a,b). These results had a large spread for all scenarios, as a consequence of differences in model characteristics and included processes, initialization methods, and the interpretation and application of model forcings (Nowicki et al., 2013b).

A limitation of these previous efforts was the use of climate forcing that could be considered as outdated by the time of the experiments. For example, the SeaRISE initiative (Sea level Response to Ice Sheet Evolution, Bindschadler et al., 2013) used

results from IPCC-AR4 scenarios while at the same time IPCC-AR5 climate simulations became available. In order to better coordinate the ice sheet modeling and climate modeling communities, the Ice Sheet Model Intercomparison Project for CMIP6 (ISMIP6) was designed to be the primary activity within the Coupled Model Intercomparison Project phase 6 (CMIP6) that focuses on the Greenland and Antarctic ice sheets (Nowicki et al., 2016).

Previous ice sheet intercomparison efforts (Pattyn et al., 2012; Pattyn et al., 2013; Bindschadler et al., 2013; Goelzer et al.,

2018) highlighted the importance of better assessing the causes of the spread in model results, and separating differences associated with model grid resolution, ice dynamics (e.g., choice of stress balance equation), physical processes included (e.g., calving, hydrofracture, and cliff failure), and initialization procedure (e.g., data assimilation, spin-up, or relaxation). While the impact of many processes and parameters can be assessed by running large ensembles (e.g., Ritz et al., 2015; Pollard et al., 2016) or using uncertainty quantification (e.g., Schlegel et al., 2015, 2018), analyzing the impact of initial conditions is more difficult.

Ice sheet models rely primarily on two methods to construct their initial state: (1) long transient simulations of ice sheet evolution since the Last Glacial Maximum or earlier, with forcing based on past climates (e.g., Huybrechts, 2002; Greve and Herzfeld, 2013; Aschwanden et al., 2013; Golledge et al., 2015) or (2) data assimilation of observed present-day conditions at a given time (e.g., Morlighem et al., 2010; Gillet-Chaulet et al., 2012; Morlighem et al., 2013; Favier et al., 2014; Arthern et al.,



2015; Cornford et al., 2015). The first captures the climate history and ensures that modeled variables are mutually consistent, but the simulated present-day ice state might differ significantly from the current observed state, which can impact the sensitivity to perturbations (Pollard and DeConto, 2012a). The second method reproduces present-day ice sheet geometry and velocity well, but does not capture past climate evolution and current trends of ice mass, due to inconsistencies between datasets (Seroussi et

al., 2011), also impacting the ice sheet response to perturbations. To combine the best of these two approaches, models using long transient spin-ups have integrated simple inverse methods to match present ice sheet geometry (Pollard and DeConto, 2012a), while models using data assimilation have run short-term relaxations to limit the initial shock caused by inconsistent datasets (Gillet-Chaulet et al., 2012). These additions are widening the spectrum of initialization methods (see also Goelzer et al., 2018).

Since ice sheets have a slow response time, their initial conditions influence their evolution for centuries to millennia. Understanding the impact of initialization methods is therefore critical for projections of sea level in the 21$^{st}$ century and beyond. The initMIP experiments were thus designed as the first part of ISMIP6, with the goal of understanding the effects of initialization procedures on model results under simplified and relatively large climate forcings. This effort is intended to enhance our understanding of the causes of variations in sea level contribution from Antarctica, but not to provide improved

estimates of sea level evolution. A previous effort, initMIP-Greenland (Goelzer et al., 2018) showed that the initial ice sheet extent has a large impact on Greenland ice sheet evolution when anomalies in surface mass balance (SMB) are applied. Here, we describe a similar effort for the Antarctic ice sheet, using simple climate anomalies applied to both the surface mass balance and to sub-ice shelf melting rates. We analyze 25 simulations from 16 international groups in order to determine the most relevant factors and to better understand the spread in projections of 21st century Antarctic ice sheet contributions to sea level.

We first describe the initMIP-Antarctica experimental design in section 2 and the participating models in section 3. In section 4, we analyze simulation results and the spread in model responses, and in section 5 we discuss these results and their implications for improving model initialization and constraining sea- level projections. We conclude with remarks relevant to future modeling efforts.

## 2 Experiments and model set-up

In this section we describe in detail the initMIP-Antarctica experiments, including model requirements and outputs. A complete documentation can be found on the ISMIP6 wiki page (http://www.climate-cryosphere.org/wiki/index.php?title=InitMIP-Antarctica).

### 2.1 Experiments description

InitMIP-Antarctica consists of an initial state, *init*, describing the initial state of the Antarctic ice sheet model, followed by three

experiments, each designed for continental-scale Antarctic simulations. Modeling groups are asked to describe the ice sheet geometry and other characteristics at the end of their initialization procedure, which is left to the discretion of each group. The following three experiments are 100-year simulations of the Antarctica ice sheet evolution under different forcing scenarios.

In *ctrl*, the control run, climate forcing is assumed to be similar to present day conditions, so atmospheric and oceanic forcings at the end of the *init* experiment are continued unchanged.

In *asmb*, the surface mass balance (SMB) anomaly experiment, atmospheric forcing evolves under a climate-change scenario associated with high greenhouse gas emissions, similar to Representative Concentration Pathway (RCP) 8.5. The prescribed anomaly is the average change in Antarctic SMB for six models: five publicly available CMIP5 RCP8.5 model




simulations (Taylor et al., 2012) with large SMB changes between 2006-2010 and 2095-2100, along with one regional model (RACMO2.1, Ligtenberg et al., 2013). As RACMO2.1 results for RCP8.5 were not available when the anomaly field was prepared, we used results for the A1B scenario, with SMB adjusted linearly to reflect the additional radiative forcing (an increase of 8.5 W/m$^2$ by 2100 in RCP8.5, compared to 6 W/m$^2$ in A1B). The RCP8.5 scenario increases precipitation by up to 50% over

the Antarctic ice sheet for some climate models (Ligtenberg et al., 2013; Palerme et al., 2016). SMB anomalies are mostly positive over the ice sheet, with a few regions seeing a negative anomaly due to increased surface runoff (Fig.1a). This anomaly is applied over the entire ice sheet.

In *abmb*, an anomaly in ocean-induced sub-ice shelf melt rates is applied under the floating ice to mimic future warming of ocean waters. It is not well understood how changes in far-field ocean conditions in global climate models transfer onto the

Antarctic continental shelf and into sub-ice shelf cavities; this is an active area of research (Nakayama et al., 2014; Asay-Davis et al., 2017; Donat-Magnin et al., 2017). We therefore apply a simple forcing anomaly equivalent to the estimated present-day melt rates under floating ice (Depoorter et al., 2013; Rignot et al., 2013). Thus, this melt rate anomaly represents a doubling of present-day estimates of melting. A different mean melt rate anomaly is specified for each of the 20 ice sheet basins, with a spatially uniform anomaly within each basin (Fig.1b). It is applied under all floating ice, including ice that ungrounds during the

experiment.

For the *asmb* and *abmb* experiments, anomalies in SMB and sub-shelf melt rates are applied in addition to the forcings used in the *init* and *ctrl* experiments. The anomalies are applied as time-dependent functions, increasing stepwise each year over the first 40 simulation years and remaining constant over the last 60 years:

$$EX(t) = EX_{ctrl} + EX_{anom} \times \frac{[t]}{40}, \qquad \text{for } 0 < t < 40 \text{ yr}$$

$$EX(t) = EX_{ctrl} + EX_{anom}; \qquad \text{for } t > 40 \text{ yr}$$

where EX($t$) is the forcing at time t, EX$_{ctrl}$ the forcing used in the *ctrl* experiment, EX$_{anom}$ the applied anomaly (Fig. 1) and [$t$] the floor function at time $t$

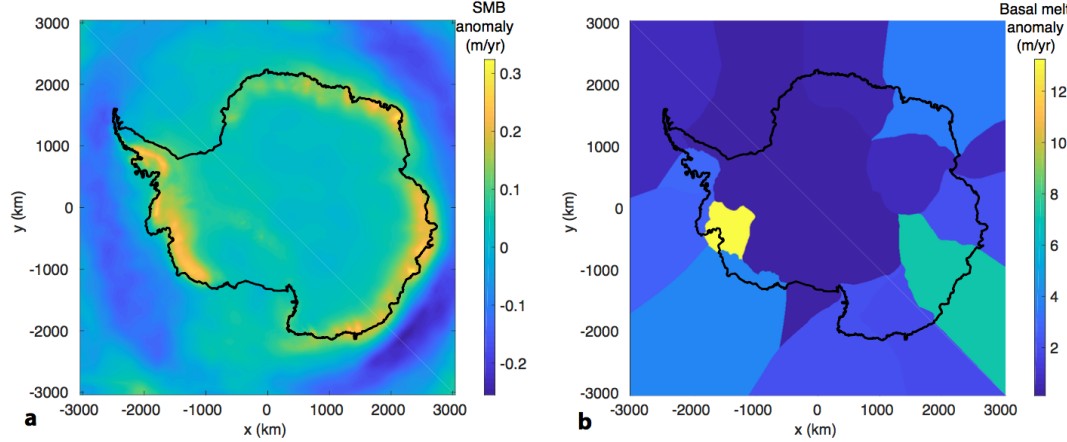

**Figure 1: (a) Surface mass balance anomaly (m/yr) for the *asmb* experiment and (b) basal melt rate anomaly (m/yr) for the *abmb* experiment. Black contours show the current Antarctic ice sheet extent.**

These forcings should not be viewed as projections of climate forcing over the coming century, but rather represent simple perturbations with relatively large changes for the purpose of assessing impacts on Antarctic ice sheet evolution.

## 2.2 Model set-up



Ice sheet models are free to use whatever initialization procedure is deemed appropriate, given model characteristics and requirements. Submitted simulations rely on long paleoclimate spin-ups, steady states, data assimilation, or a combination of these methods. There is no constraint or suggestion on forcing datasets (including SMB and sub-shelf melt rates) or on specific physical processes and parameterizations (e.g., basal sliding laws, ice rheology, and stress balance approximation). The

initialization time varies among models but is near the beginning of the 21$^{st}$ century.

Previous multi-model ice sheet studies (Bindschadler et al., 2013; Nowicki et al., 2013a,b) showed the difficulty of separating the effects of initial conditions, physical processes, and external forcings. In order to better analyze the links between initial conditions and external forcings, we impose several modeling constraints. Models are required to model floating ice shelves and grounding line dynamics as changes in ice shelves significantly impacted the evolution of West Antarctica in the

past decades. The exact procedure to simulate these processes, however, is left at the discretion of the modeling groups. Ice sheet models should apply the provided SMB anomalies without adjusting for geometric changes in forward experiments (i.e., surface-elevation feedback). Similarly, they should apply the basal melt rate anomaly under floating ice as it evolves over time. Finally, bedrock elevation adjustment, ice shelf hydrofracturing, and ice cliff failure should not be included.

### 2.3 Model outputs

Modeling groups were requested to report simulation results using a standard output format. Table A1 lists the required outputs, including both scalar and 2D variables. Scalar variables are values describing the entire ice sheet (e.g., ice mass, ice mass above flotation, and area-integrated SMB and basal melting). Three kinds of 2D outputs are requested. State variables (e.g., ice velocity and thickness) are snapshots reported at a given time; flux variables are reported as averages over a given period; and constant variables do not change with time.

Scalar outputs are provided for each simulation year and corrected for area distortion due to the projection (e.g., polar stereographic), while 2D variables (e.g., ice thickness, surface temperature, and basal drag) are reported every 5 years. For 2D variables, results are reported on prescribed regular grids to help achieve a consistent analysis. These grids are defined on a polar stereographic projection with standard parallel 71°S and central meridian 0°E. Modelers are free to use one of the six prescribed grids with the resolution closest to their native resolution. All outputs are then regridded using a conservative interpolation

scheme (Jones, 1999) onto an 8 km grid that is used for the analysis. The output grids are similar to the grids used to provide the SMB and basal melt anomalies.

### 3 Participating models

Sixteen modeling centers participated to the initMIP-Antarctica effort and submitted 25 simulations; each model performed the whole suite of experiments. The list of modeling centers is shown in Table 1. Table 2 lists the main characteristics of each

simulation, including the stress balance approximation, grid resolution, initialization procedure, initial year, and external forcing. More details on individual models and initialization procedures can be found in Appendix B.

A majority of models uses the finite difference method, with two models based on finite volumes, two based on the finite element method, and two based on a combination of finite element and finite volume. Two simulations use the Shelfy Stream Approximation (SSA, MacAyeal, 1989), three use L1L2 (i.e.. depth-integrated higher-order) approximations (Hindmarsh, 2004),

and two use a 3D higher-order approximation (Pattyn, 2003). The other models use a combination of the Shallow Ice Approximation (SIA, Hutter, 1983) and SSA, either combining SIA for the grounded ice with SSA for the floating ice, or using SSA as a sliding law and SIA for the internal deformation (Bueler and Brown, 2009). The grid resolution ranges from 4 to 32 km





for models based on fixed regular grids, while models using adaptive grid refinement are able to use resolutions as low as 0.5 km in grounding zones.

| Contributors | Group ID | Ice flow model | Group |
|---|---|---|---|
| Nicholas Golledge<br>Daniel Lowry | ARC | PISM | Antarctic Research Centre,<br>Victoria University of Wellington, New-Zealand |
| Thomas Kleiner<br>Johannes Sutter<br>Angelika Humbert | AWI | PISM | Alfred Wegener Institute for Polar and Marine Research,<br>Bremerhaven, Germany |
| Stephen Cornford | CPOM | BISICLES | Swansea University, United Kingdom |
| Christian Rodehacke | DMI | PISM | Danish Meteorological Institute, Denmark |
| Matthew Hoffman<br>Tong Zhang<br>Stephen Price | DOE | MALI | Los Alamos National Laboratory, USA |
| Julien Brondex<br>Fabien Gillet-Chaulet | IGE | Elmer/Ice | Institut des Géosciences de l'Environnement, France |
| Ralf Greve | ILTS | SICOPOLIS | Institute of Low Temperature Science,<br>Hokkaido University, Sapporo, Japan |
| Heiko Goelzer<br>Thomas Reerink<br>Roderik van de Wal | IMAU | IMAUICE | Institute for Marine and Atmospheric Research,<br>Utrecht, The Netherlands |
| Nicole Schlegel<br>Helene Seroussi | JPL | ISSM | Jet Propulsion Laboratory, California Institute of Technology, Pasadena, USA |
| Christophe Dumas<br>Aurelien Quiquet | LSCE | Grisli | Laboratoire des Sciences du Climat et de l'Environnement<br>Université Paris-Saclay, France |
| Gunter Leguy<br>William Lipscomb | NCAR | CISM | National Center for Atmospheric Research, Boulder, CO, USA |
| Torsten Albrecht<br>Matthias Mengel<br>Ronja Reese<br>Ricarda Winkelmann | PIK | PISM | Potsdam Institute for Climate Impact Research, Germany |
| David Pollard | PSU | PSU | Earth and Environmental Systems Institute, Pennsylvania<br>State University, University Park, PA, USA |
| Mathieu Morlighem<br>Helene Seroussi | UCIJPL | ISSM | University of California, Irvine, USA<br>Jet Propulsion Laboratory, California Institute of Technology, Pasadena, USA |
| Frank Pattyn<br>Sainan Sun | ULB | f.ETISh | Université libre de Bruxelles, Belgium |
| Jonas Van Breedam<br>Philippe Huybrechts | VUB | AISMPALEO | Vrije Universiteit Brussel, Belgium |

**Table 1: List of participants, modeling groups and ice flow models in ISMIP6 initMIP-Antarctica.**





The initialization methods cover the spectrum of procedures used in the ice sheet modeling community. Fourteen simulations are based on a paleoclimate spin-up with forcings reproducing the evolution of climate during the simulated period, and four of these simulations have a targeted ice sheet geometry at the end of their run, similar to the method described in Pollard and DeConto (2012a). Four models are based on a steady-state equilibrium in which the model is run for an extended period of time, until the

ice sheet becomes close to a steady-state equilibrium, with two model also including present-day geometry as a target (Pollard and DeConto, 2012a). The remaining seven initializations are based on data assimilation, with three models also including a short relaxation period after the data assimilation to limit the impact of inconsistent datasets (Seroussi et al., 2011; Gillet-Chaulet et al., 2012).

For the external SMB forcing, models use output from RACMO2 (Lenaerts et al., 2012), RACMO2.3 (van Wessem et al.,

2014), MAR (Gallée et al., 2013), ERA Interim (Dee et al., 2011), or Arthern et al. (2006). Five simulations use a positive-degree-day scheme (PDD, Reeh, 1991). These choices generate relatively similar initial SMB (see section 4). For sub-shelf melting, three simulations do not apply any melt rate. Four others apply values estimated from remote sensing, extrapolated to regions that unground during the simulation. Most models apply a parameterization that depends linearly (Martin et al., 2011, 8 simulations) or quadratically (DeConto and Pollard, 2016, 4 simulations) on the ocean thermal forcing. Three simulations adjust

the melt rate using an observed thickness target, and the remaining three simulations use the new PICO parameterization (Reese et al., 2018).

Most models include a moving ice front, but five simulations have a fixed ice front. Ice front migration is based on strain rate in most cases (Levermann et al. (2012), 10 simulations). Some models use ice flux divergence and accumulated damage at the ice front (Pollard et al. (2015), 3 simulations), some have ice-front retreat based on a threshold ice thickness (4 simulations),

while the others have retreat only where the ice melts completely (3 simulations).



| Model name | Numerics | Stress balance | Resolution (km) | Initialization | Initial Year | Initial SMB | Initial basal melt | Melt in partially floating cells | Ice Front |
|---|---|---|---|---|---|---|---|---|---|
| ARC_PISM1 | FD | Hybrid | 16 | SP | 2000 | RA2 | 0 | No | StR |
| ARC_PISM2 | FD | Hybrid | 16 | SP | 2000 | RA2 | 0 | Sub-Grid | StR |
| ARC_PISM3 | FD | Hybrid | 16 | SP | 2000 | RA2 | Lin | No | StR |
| ARC_PISM4 | FD | Hybrid | 16 | SP | 2000 | RA2 | Lin | Sub-Grid | StR |
| AWI_PISM1Eq | FD | Hybrid | 16 | Eq | 1979-2011 | RA2.3 | Quad | No | StR |
| AWI_PISM1Pal | FD | Hybrid | 16 | SP | 1979-2011 | RA2.3 | Quad | No | StR |
| CPOM_BISICLES_A | FV | L1L2 | 0.5-8 | DA+ | 2010 | Art | SS | No | RO |
| CPOM_BISICLES_B | FV | L1L2 | 0.5-8 | DA+ | 2010 | Art | SS* | No | RO |
| DMI_PISM0 | FD | Hybrid | 16 | SP | 1979-2012 | ERA | Lin | No | StR |
| DMI_PISM1 | FD | Hybrid | 16 | SP | 1979-2012 | ERA | Lin | No | StR |
| DOE_MALI | FE/FV | HO | 2-20 | DA+ | 2007 | RA2 | Obs | No | Fix |
| IGE_ELMER | FE | SSA | 1-50 | DA+ | 2000 | MAR | Lin | No | Fix |
| ILTS_SICIPOLIS1 | FD | Hybrid | 8 | SP | 1990 | PDD | Lin | N/A | MH |
| ILTS_SICIPOLIS2 | FD | Hybrid | 8 | SP | 1990 | PDD | Lin | N/A | MH |
| IMAU_IMAUICE32 | FD | Hybrid | 32 | Eq | 1990 | RA2.3 | 0 | No | Fix |
| JPL1_ISSM | FE | SSA | 1-50 | DA | 2007 | RA2 | Obs | Sub-Grid | Fix |
| LSCE_GRISLI | FD | Hybrid | 16 | SP+ | 2000 | MAR | SS | N/A | MH |
| NCAR_CISM | FE/FV | L1L2 | 4 | SP+ | 1979-2016 | RA2 | Obs | Floating condition | RO |
| PIK_PISM3PAL | FD | Hybrid | 16 | SP+ | 1986-2005 | RA2.3 | PICO | Sub-Grid | StR |
| PIK_PISM4EQUI | FD | Hybrid | 8 | Eq+ | 1986-2005 | RA2.3 | PICO | Sub-Grid | StR |
| PSU_EQNOMEC | FD | Hybrid | 16 | Eq+ | 1979-2010 | PDD | Quad | Sub-Grid | Div |
| PSU_GLNOMEC | FD | Hybrid | 16 | SP+ | 1979-2010 | PDD | Quad | Sub-Grid | Div |
| UCIJPL_ISSM | FE | HO | 3-50 | DA | 2007 | RA2 | Obs | Sub-Grid | Fix |
| ULB_FETISH1 | FD | Hybrid | 16 | DA* | 1979-2014 | RA2.3 | PICO | N/A | Div |
| VUB_AISMPALEO | FD | SIA+SSA | 20 | SP | 2000 | PDD | Lin | N/A | MH |

**Table 2: List of initMIP-Antarctica simulations and main model characteristics. Initialization methods are: Spin-up (SP), Spin-up with target values for the ice thickness (SP+, see Pollard and DeConto, 2012a), Data Assimilation (DA), Data Assimilation with short relaxation (DA+), Data Assimilation of ice geometry (DA*), Equilibrium state (Eq), and Equilibrium state with target values for the ice thickness (Eq+). Initial SMB is derived from: RACMO2 (RA2, Lenaerts et al., 2012), RACMO2.3 (RA2.3, van Wessem et al., 2014), MAR (Gallée et al., 2013), ERA Interim (ERA, Dee et al., 2011), Arthern et al. (2006) (Art), and positive-degree-day schemes (PDD, Reeh, 1991). Basal melt rates are based on zero melting (0), linear function of thermal forcing (Lin, Martin et al., 2011), quadratic function of thermal forcing (Quad, DeConto and Pollard, 2016), melt rates estimated from observations (Obs, Rignot et al., 2013; Depoorter et al., 2013), ice shelves thickness target (SS), ice shelves thickness target with no refreezing (SS ), and the PICO parameterization (Reese et al., 2018). Models that have partially floating cells at the grounding line apply melting using a sub-grid scheme (Sub-Grid), a floatation condition or no melt at all in their partially floating cells. Ice front migration schemes are based on strain rate (StR, Levermann et al., 2012), retreat only (RO), fixed front (Fix), minimum thickness height (MH) and divergence and accumulated damage (Div, Pollard et al., 2015). Further details on all the models are given in Appendix B.**





## 4 Results

### 4.1 *init* experiment

Each model reports initial ice sheet conditions at the end of the initialization procedure (*init*).The total ice-covered area varies
between $1.35 \times 10^7$ km$^2$ and $1.50 \times 10^7$ km$^2$, a range of only 10.5% among models. The ice shelf extent, on the other hand, varies

5  significantly among models, from $0.92 \times 10^6$ km$^2$ to $2.51 \times 10^6$ km$^2$, a range of 6.4% to 16.7% of the total ice-covered area. Figure
summarizes the initial extent of all models. Some models have ice shelves hundreds of kilometers upstream or downstream of
their current observed location. Although models generally agree on the location of the three largest ice shelves (Ross, Ronne-
Filchner and Amery), the location and extent of smaller shelves varies widely, including in the Amundsen and Bellingshausen
Sea sectors. The initial ice mass above flotation varies from $1.79 \times 10^7$ Gt to $2.47 \times 10^7$ Gt (between 49.4 and 68.1 m of SLE),

while the total ice mass varies from $2.11 \times 10^7$ to $2.56 \times 10^7$ Gt, in part because of the large discrepancy in ice shelf extent. Table
C1 details the main scalar variables in *init* for all simulations.

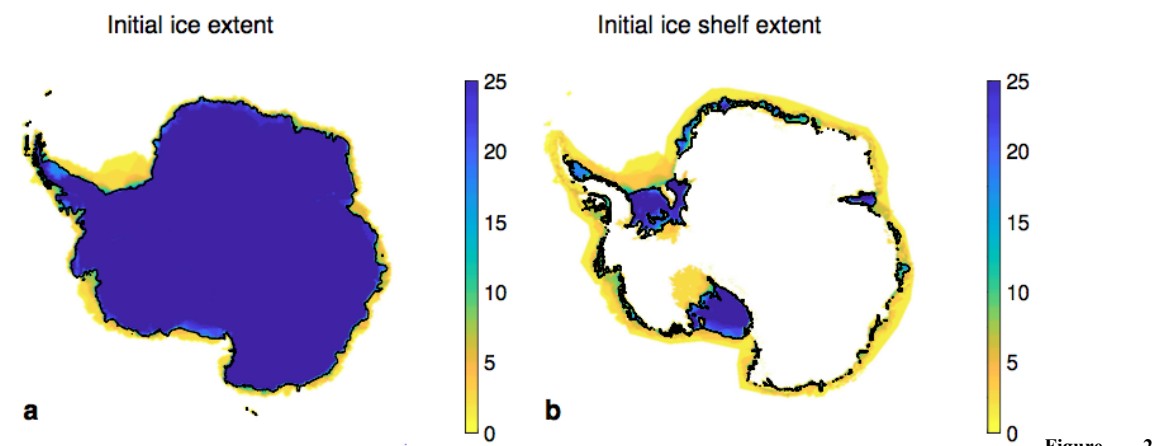

**Figure 2:**
**Initial extent of ice-covered areas and ice shelves for all participating models. All contributions are regridded onto an 8 km standard
grid. Figures indicate how many models include ice (ab), or floating ice (b), in each grid cell. Black lines show the observed ice extent**
**(a) and ice shelf extent (b) from Bedmap2 (Fretwell et al., 2013).**

The ability of models to reproduce the characteristics of the present-day ice sheet depends on their initialization procedure.
The root mean square error (RMSE) between observed (Fretwell et al., 2013) and modeled ice thickness varies between 91.2 m
and 422.3 m, with generally smaller errors (between 91.2 and 320.8 m) for models using data assimilation or present-day
geometry as a target in their initialization, and larger errors (between 160.0 m and 422.3 m) for models using spin-up, steady

state or long relaxation procedures without a geometry target (Fig. 3a). The RMSE between observed (Rignot et al., 2011a) and
modeled surface velocity (Fig. 3b) also has a large spread among models, varying from 47.5 m/yr to 308 m/yr. These values are
significantly affected by the inclusion of observed surface velocities during the initialization procedure: the RMSE in surface
speed varies from 47.5 m/yr to 94.5 m/yr for models including data assimilation of surface velocities, and from 116 m/yr to 308
m/yr for the other models. Most of these errors are caused by large discrepancies in ice shelves and a few fast-flowing ice

streams: the RMSE for the logarithm of the speed, which emphasizes the slower moving regions, varies only between 0.62 and
1.51 (Fig. 3c), or three times less than the RMSE of the speed. These errors are in part affected by the exact year of the
initialization procedure, as observations of velocity and thickness are not acquired at the same time. Thickness and velocity
variability is small, however, compared to the discrepancies between observations and models.





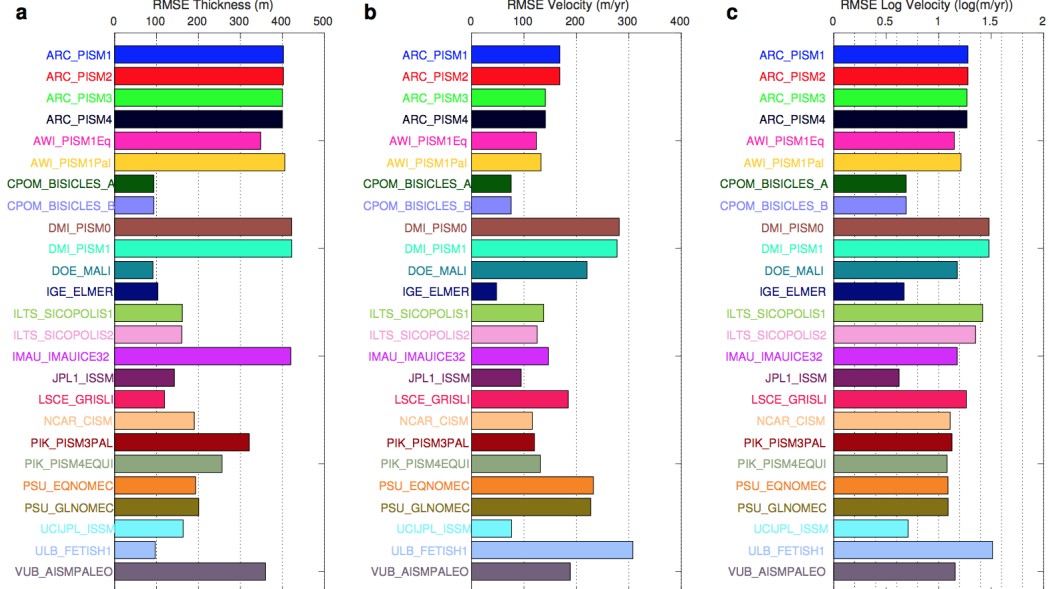

**Figure 3: Root Mean Square Error (RMSE) of modeled initial conditions compared to observations for (a) initial ice thickness (m), (b) initial ice surface velocity(m/yr) over the ice sheet and ice shelf and (c) the logarithm of the initial ice surface velocity (log(m/yr)). Please note, that the model-color relationship used in this figure is applied in all subsequent figures.**

Area-integrated external forcings (SMB and basal melt) also differ substantially among the models (see Table C1). The total

initial SMB varies from 2015 Gt/yr to 3430 Gt/yr, depending on the origin of the SMB forcing (see Table 2) and the extent of the

ice-covered areas. The total initial basal melt varies from 0 to 2470 Gt/yr, with seven models having values of less than 150

Gt/yr, while remote sensing estimates of total Antarctic basal melt are ~1400 Gt/yr (Rignot et al., 2013; Depoorter et al., 2013).

Similar to the SMB forcing, these differences result from the chosen melting parameterization (Table 2) and the geometry of ice

shelves.

**4.2 *ctrl* experiment**

Representing the current state of the ice sheet does not guarantee that the current trends in ice sheet changes are correctly
captured, which is what eventually matters in sea level rise projections. In the *ctrl* experiment, the Antarctic ice sheet evolves
under a constant climate for 100 years. The total change of ice mass above flotation varies from a loss of 60,500 Gt to a gain of
88,100 Gt (i.e., 243 mm of SLE drop to 167 mm of SLE rise; see Fig. 4a and Table B2), with mass loss in 8 simulations and gain
in 17 simulations. This absolute change in mass above flotation represents less than 0.42% of the initial volume in all cases,
highlighting the accuracy required to calculate the Antarctic evolution for sea level projections. A spread of results is observed
for all initialization methods and model resolutions. Eleven models have an absolute change lower than 20 mm, ten have an
absolute change above 80 mm, and four an absolute change between 20 and 80 mm. All the models initialized with a steady-state
equilibrium but one, have a sea level change lower than 20 mm, while all the models using data assimilation to determine their
initial conditions but one, have a sea level change above 80 mm. The models based on a paleo-climate spin-up  have a large
spread of sea level change in the *ctrl* experiment, and are present in all categories. The number of models in each category is,
however, relatively small to draw definitive conclusions.





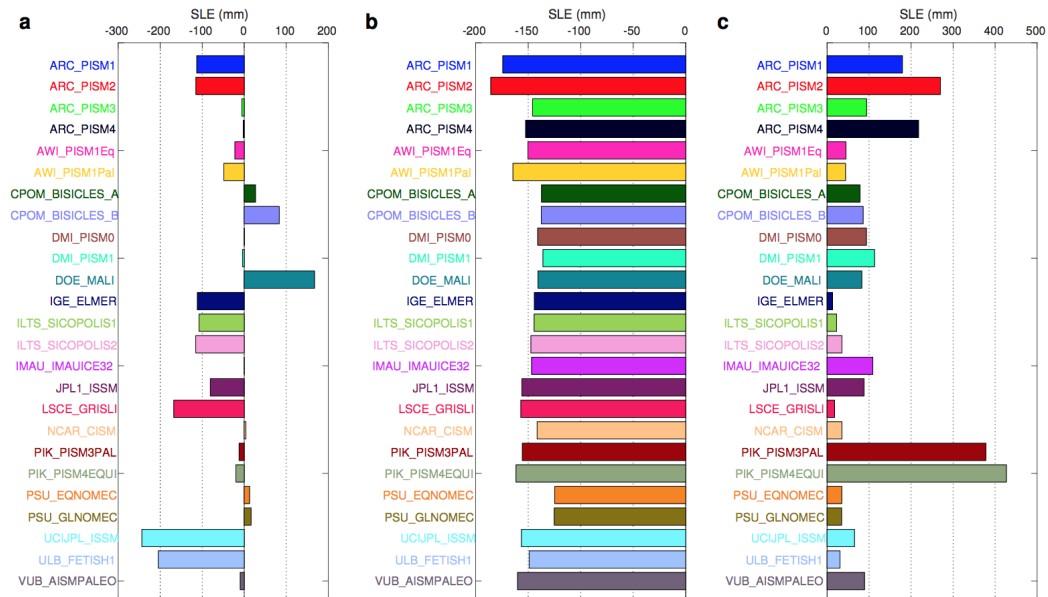

**Figure 4:**
**Antarctic contribution to sea level (mm of sea level equivalent). (a)** *ctrl* **experiment, (b) difference between** *asmb* **and** *ctrl* **experiments, and (c) difference between** *abmb* **and** *ctrl* **experiments.**

Figure 5 shows the spatial patterns of thickness and depth-average horizontal ice speed for the *ctrl* experiment. Regridded
5  results on the 8 km standard grid are used to compute mean changes and standard deviation for these two variables. Results are
reported only where at least five simulations have ice at a given grid point. Maps of thickness and velocity change during the *ctrl*
experiment show that the signals are larger along the coast than in the interior of the continent, and larger in West Antarctica
compared to East Antarctica. The mean thickness change over the ice sheet, averaged over all models, is equal to 1.2 m in 100
years. The standard deviation is calculated for each grid cell of the 8 km standard grid based on the number of models reporting
10  results in each cell, and excluding cells where less than 5 models simulate ice. The standard deviation is much larger than the
mean changes in many places, with an average value over the simulated area of 14.8 m. Substantial thickening and thinning
(especially of ice shelves) compensate each other, leading to a small average change but large standard deviation. Similarly, the
average velocity change is small, with a value of -1.9 m/yr, but the standard deviation is 27.4 m/yr. Some models have large
accelerations in key regions, while others have large slowdowns. Regions with the largest spread in model thickness and velocity
15  changes are generally similar.





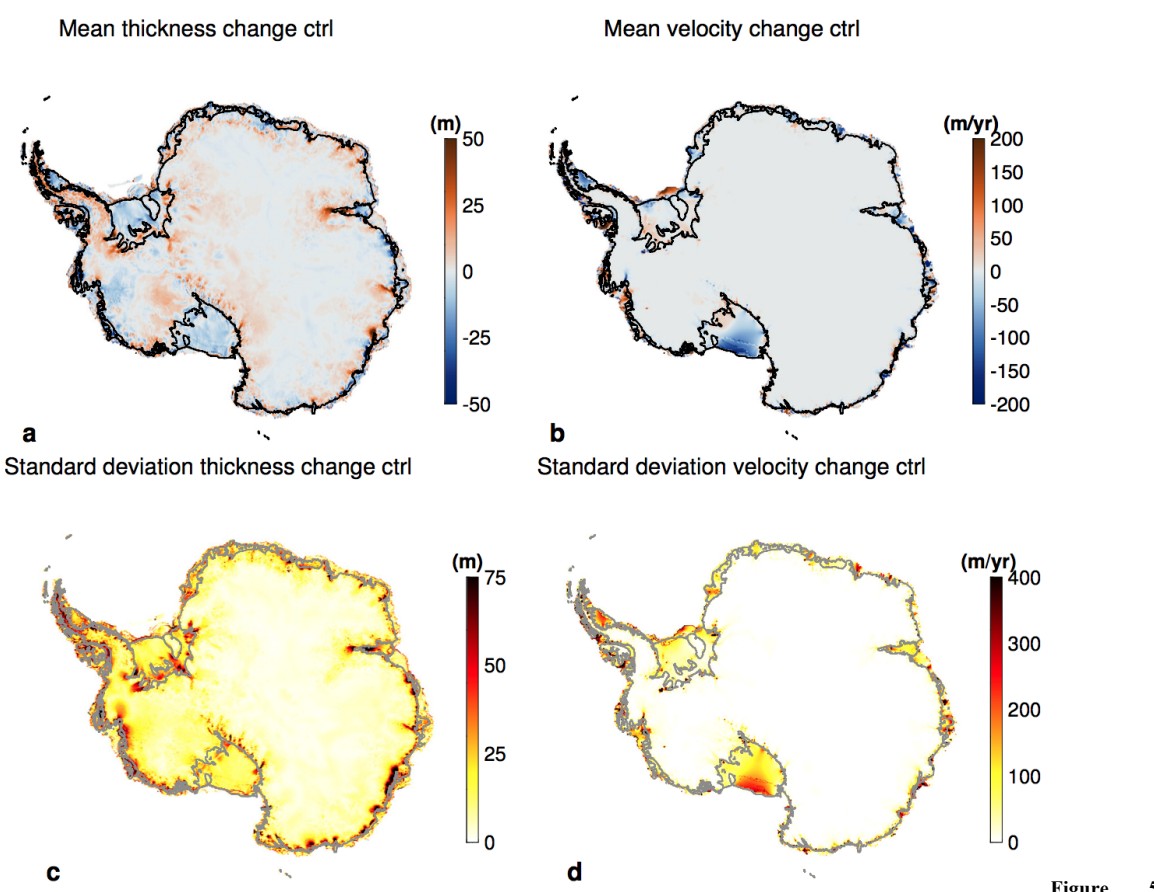

**Figure 5:**
**Mean (a and b) and standard deviation (c and d) of the change in ice thickness (a and c, in m) and depth-averaged horizontal velocity (b and d, in m/yr) between the beginning and end of the *ctrl* experiment. Black (a and c) or grey (b and d) lines show the observed current ice front and grounding line positions.**

5    The ice extent is relatively stable in all *ctrl* simulations, with less than 1.3% change in the most sensitive simulations. Some simulations, however, have large changes in ice shelf extent, ranging from a reduction of 13% to an increase of 14%. The area-integrated SMB varies by up to 6% for the simulations that experience the largest change in SMB (Fig. 6b). The area-integrated basal melting varies by more than 5% for 15 models, with a maximum change of 29%, in response to changes in ice shelf extent and thickness (Fig. 6c).

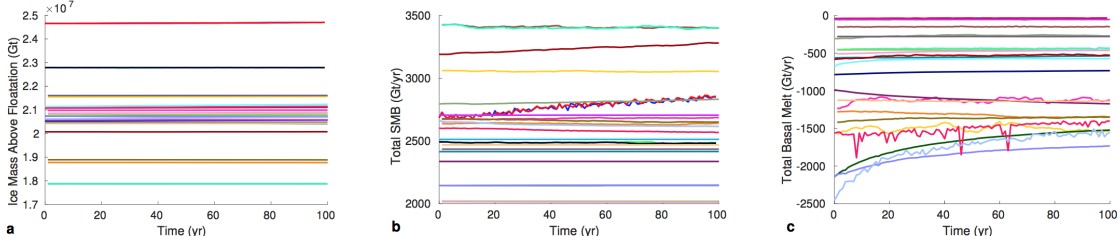

10   **Figure 6: Evolution of Antarctic ice sheet mass above floatation and external forcings in the *ctrl* experiment. (a) Total mass of ice above floatation (Gt), (b) total SMB applied at the ice surface (Gt/yr), and (c) total basal melting rate (Gt/yr).**

**4.3 *asmb* experiment**





In the *asmb* experiment, an SMB anomaly (Fig. 1a) is added to the SMB used in the *ctrl* experiment. This anomaly leads to an increase in ice mass above flotation compared to *ctrl*, with the mass gain ranging from $4.51 \times 10^4$ Gt to $6.72 \times 10^4$ Gt (125–186 mm decrease in SLE, see Fig. 4b). The differences among models (Fig. 7a,b) are linked to the extent of the ice-covered areas, as well as ice shelves extent. For most models there is a small increase in grounded area, as some floating areas near grounding

lines thicken and reground due to the positive SMB anomaly.

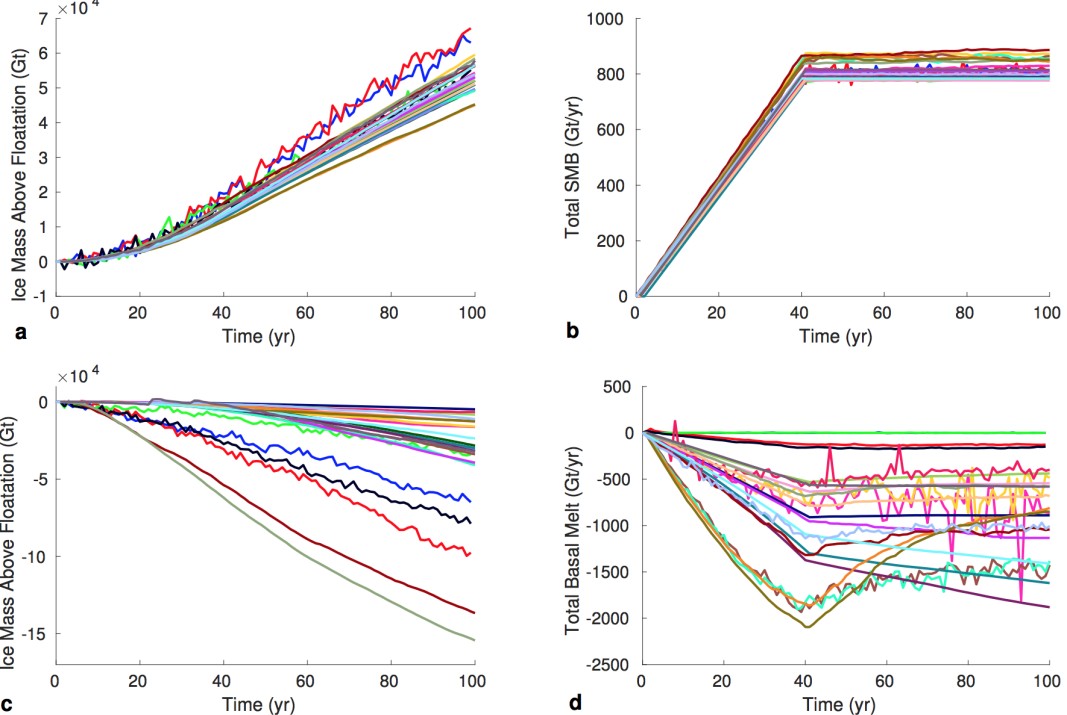

**Figure 7: Evolution of the Antarctic ice sheet and external forcings in the *asmb* (upper row) and *abmb* (lower row) experiments compared to the *ctrl* experiment. Total amount of ice above floatation for *asmb* minus *ctrl* (a) and *abmb* minus *ctrl* (c), in Gt. Evolution of SMB applied at the ice surface for *asmb* minus *ctrl* (b, in Gt/yr) and total basal melting rate applied in *abmb* minus *ctrl* (d, in Gt/yr).**

Figure 8 shows the mean and standard deviation of the impact of this SMB anomaly on the ice thickness and depth-averaged horizontal velocity. Figure 8 is similar to Fig. 5, but for the difference between the end of the *asmb* experiment and the end of the *ctrl* experiment. As expected from the SMB anomaly spatial pattern (Fig. 1a), there is a thickening of 3.6 m on average over Antarctica, with the largest changes happening along the West Antarctic coasts and the Antarctic Peninsula (Fig. 8a). The standard deviation map (Fig. 8c) shows that model differences are again concentrated along the West Antarctica coast and on the

Peninsula. The average standard deviation over the continent is 5.2 m for this anomaly. The SMB anomaly has a small impact on ice dynamics, as shown in Fig. 8b, with an average speed increase of 1.5 m/yr over 100 years and a standard deviation of 17.6 m/yr. Regions where models disagree are similar to those for the *ctrl* experiment. Fig. 9 compares for each model the difference in mass between the end of the *asmb* experiment and the end of the *ctrl* experiment with the cumulative SMB anomaly of experiment *asmb* integrated over the entire ice sheet. It confirms that the additional surface mass balance is the primary cause of

mass change: the SMB anomaly explains between 97 and 130% of the total mass change. The difference between the cumulative SMB anomaly and the change in mass is caused by thicker and faster ice (see Fig. 8) that increase the calving flux, as well as feedbacks on ice shelf basal melt.



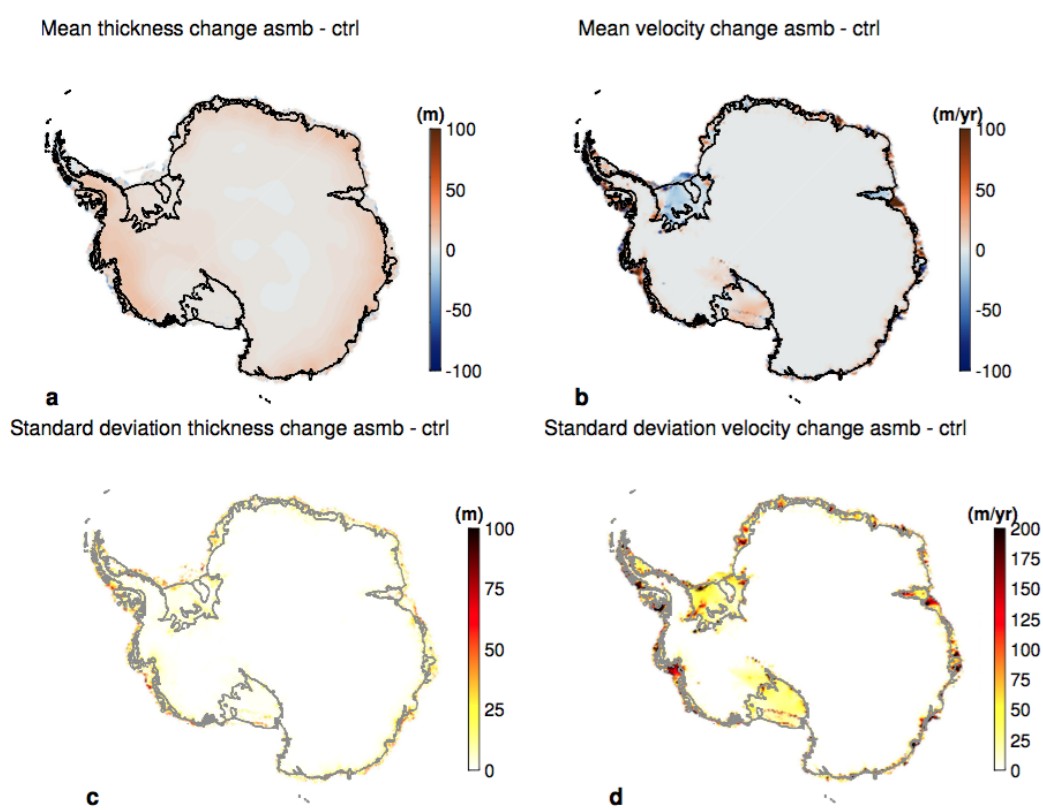

**Figure 8: Mean (a and b) and standard deviation (c and d) of the ice thickness (a and c, in m) and depth-averaged horizontal velocity (b and d, in m/yr) between the end of the *asmb* experiment and the end of the *ctrl* experiment. Black (a and b) or grey (c and d) lines show the current observed ice front and grounding line positions.**

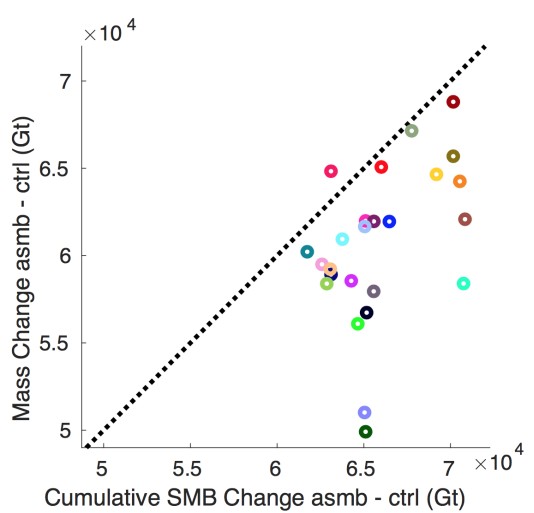

**Figure 9: Difference in mass (Gt) between the end of the *asmb* experiment and the end of the *ctrl* experiment with the cumulative SMB anomaly (Gt) of experiment *asmb* integrated over the entire ice sheet for the 25 simulations. Black dashed line shows mass change equal to cumulative SMB anomaly change.**



### 4.4 *abmb* experiment

In the *abmb* experiment, an anomaly is applied to the basal melting rate of floating ice shelves, in addition to the basal melting used in the *ctrl* experiment. The basal melt anomaly is uniform within each region (see Fig. 1b) and largest in the Amundsen Sea, where an additional ocean induced melt of 13.2 m/yr is applied. This additional melting leads to a thinning of ice shelves, a

reduction of the buttressing they provide to grounded ice, an acceleration of the ice streams feeding the shelves, and a retreat of grounding lines. However, unlike what is observed for the *asmb* experiment, the *abmb* response varies significantly among models.

Differences can be attributed in part to different treatments of basal melt in model cells near the grounding line. Some models have no melting in partly floating cells, others apply melt in partly floating cells based on the fraction of floating area,

and one model applies melt over the entire cell if it satisfies a flotation criterion (see Table 2). The spread in ice mass loss above flotation compared to the end of the *ctrl* experiment varies by two orders of magnitude, from $4.7 \times 10^3$ Gt to $1.5 \times 10^5$ Gt (or 13–427 mm of SLE; see Fig. 4c and Table B2)., even though the additional melt is applied only to floating ice, and therefore does not contribute directly to sea level rise. The grounded area is reduced for all the models (between 0.10% and 1.7% reduction) as grounding lines retreat. The change in ice shelf extent varies from a reduction of 25% to an increase of 12%, as some ice shelves

calve during this experiment, depending on the choice made for ice front evolution (see Table 2).

Fig. 10 shows that the mean and standard deviation for the ice thickness and depth-averaged velocity changes are concentrated on the ice shelves and near grounding lines. Ice thinning is 10.7 m on average, and the standard deviation is 12.4 m. The dynamic impact of such changes is not limited to the ice shelves but propagates upstream of the grounding line, especially in the Amundsen Sea Basin, where the largest anomalies are applied. The Ross and Filchner-Ronne ice shelves have acceleration

near the grounding line, but also a slowdown near the ice front. The mean velocity change over the ice sheet is a small slowdown of 3.3 m/yr; this signal is small compared to the standard deviation of 29.6 m/yr. Regions where models show a large spread of thickness and velocity changes are different from the *ctrl* and *asmb* simulations. Large deviations among models extend upstream from the present-day grounding line and over the ice streams feeding the ice shelf, reflecting different model responses to this oceanic forcing. Figure 11 compares for each model the difference in mass between the end of the *abmb* experiment and

the end of the *ctrl* experiment with the cumulative basal melt anomaly of experiment *abmb* integrated over the entire ice sheet. It shows that the additional basal melt only accounts for a fraction of the mass change: the basal melt anomaly explains between 5 and 125% of the total mass change. The difference between the cumulative basal melt anomaly and the change in mass is mainly caused by thinner and slower ice shelves (see Fig. 10) that reduce the calving flux.



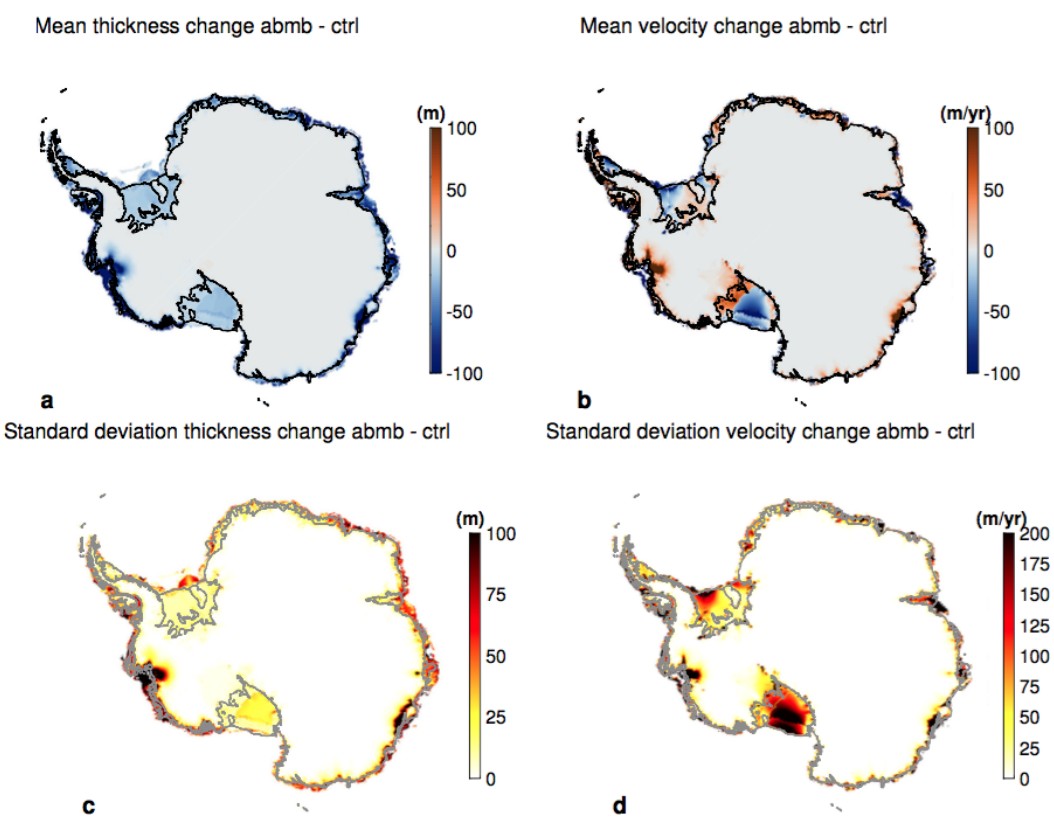

**Figure 10: Mean (a and b) and standard deviation (c and d) of the change in ice thickness (a and c, in m) and depth-averaged horizontal velocity (b and d, in m/yr) between the end of the *abmb* experiment and the end of the *ctrl* experiment. Black (a and b) or grey (c and d) lines show the current observed ice front and grounding line positions.**

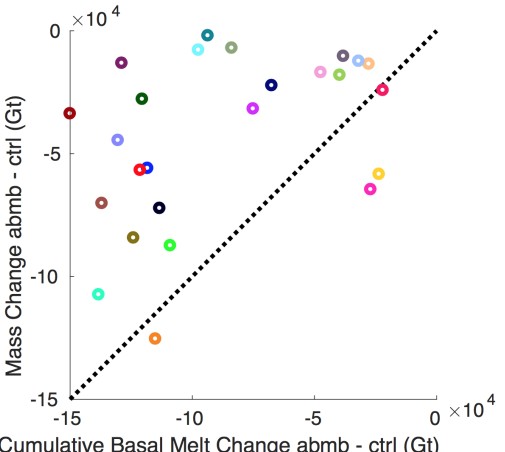





**Figure 11: Difference in mass (Gt) between the end of the *abmb* experiment and the end of the *ctrl* experiment and the cumulative basal melt anomaly (Gt) of experiment *abmb* integrated over the entire ice sheet for the 25 simulations. Black dashed line shows mass change equal to cumulative basal melt anomaly change.**

## 5 Discussion

The initMIP-Antarctica experiments are designed to analyze the impact of ice sheet model initial conditions on the evolution of the Antarctic ice sheet and its response to simple climate forcings. For this exercise, 16 groups submitted 25 simulations, more than four times the number of Antarctic simulations submitted for the SeaRISE project (Bindschadler et al., 2013), highlighting the importance and the fast evolution of this research field (Pattyn et al., 2017). The simulations represent a large diversity of initialization methods, forcing datasets, and model parameters, and the results show a large spread in the mass balance and

dynamic evolution of this ice sheet in century-scale simulations.

The initial ice volume above floatation varies from $1.8–2.5 \times 10^7$ Gt, or almost 32%, which is much larger than the spread of about 8% in SeaRISE (Nowicki et al. (2013b)). This is not surprising given the larger number of model contributions. On the other hand, the largest drifts in the *ctrl* experiment are reduced compared to the SeaRISE project. For initMIP-Antarctica, the *ctrl* sea level contribution varies between -243 mm and +167 mm of sea level equivalent for the 25 simulations of ISMIP6, while its

evolution varied between -256 mm and +1 mm over the first 100 years for the 6 simulations of SeaRISE. Specifically, four models participated in both SeaRISE and initMIP-Antarctica, and the large drift that two of them experienced in SeaRISE has been reduced in the initMIP-Antarctica *ctrl* experiment.

The *asmb* and *abmb* experiments are designed to analyze the ice sheet response to simple anomalies in SMB and basal melting under the ice shelves. Unlike initMIP-Greenland, where Goelzer et al. (2018) observed a large spread of 118% in the

responses in the *asmb* experiment, the response to the SMB anomaly in initMIP-Antarctica is similar among all the models, with a 39% variation in the response to this anomaly between the models. The differences can be attributed to the larger spread in initial ice sheet extent and the pattern of the SMB anomaly in initMIP-Greenland. In Greenland, large ablation rates are applied at the ice sheet periphery, leading to significant ice loss for the models with the largest initial extents (Goelzer et al., 2018). The Antarctic SMB anomaly has less spatial variability, and the initial extent of the ice sheet is closer for the different simulations,

which leads to more consistent responses to this perturbation.

While the response to the SMB anomaly has limited variations among models, the impact of the basal melting anomaly varies significantly among models, with a spread in sea level contribution from 13 mm to more than 400 mm. Several factors explain the wide range of *abmb* responses. First, models vary in their treatment of basal melting near the grounding line. Elements and grid cells crossed by the grounding line are considered partly floating. Some models have no melting in partly

floating cells, others apply melt in partly floating cells based on the fraction of floating area, and one model applies melt over the entire cell if it satisfies a flotation criterion (see Table 2). These different treatments can have a significant impact on grounding line evolution as highlighted by previous studies (Arthern and Williams, 2017; Seroussi and Morlighem, 2018). This is especially important for continental- scale simulations that have a resolution varying between several km and several tens of km, as is the case in initMIP-Antarctica. The four largest sea level contributions in the *abmb* experiment (>200 mm) come from four models

that apply sub-grid melt in partly floating cells and have a resolution of 8 km or coarser (see Table 2 and Table B2). Additionally, two of these models were run without (ARC_PISM1 and ARC_PISM3 and with (ARC_PISM2 and ARC_PISM4) a sub-grid melt scheme in partially floating cells (see Table 2), which resulted in an additional sea level rise of 90 and 124 mm when the sub-grid melt scheme is used.

Second, the total ice shelf extent varies by more than 100% among the different models, and their extent within different

basins also varies significantly (see Fig. 2 and Table B1). As the basal melting anomaly is applied only under floating ice, the





spatial extent and amount of the applied anomaly therefore varies significantly from one model to the next. Ice shelf extent also changes during the *ctrl* and *abmb* experiments, so that the applied melt anomaly evolves differently between the simulations. As shown in Fig. 12, floating ice areas stay relatively constant in some models, increase because of grounding line retreat in others, and decrease as ice shelves thin significantly and calve in the remaining ones.

Third, while the SMB applied in *init* and *ctrl* is relatively similar among the different models, the basal melting varies from zero melt to 2140 Gt/yr. The latter value is about 50% larger than values derived from remote sensing observations (Rignot et al., 2013; Depoorter et al., 2013) (see Fig. 7). The applied basal melting anomaly therefore represents about half the initial basal melting for some models but a drastic increase for others. The impact on ice shelf thickness evolution and dynamic response is therefore very different, as shown on Fig. 10.

Finally, surface-elevation feedback processes were not allowed in *asmb*, ensuring that a similar SMB anomaly was applied by all models at a given location. In *abmb*, no such constraint was prescribed, which introduces feedbacks between ice shelf and basal melting for some parameterizations. For example, if an ice shelf thins and the grounding line retreats in a given model, the newly floating ice experiences basal melting that can drive further thinning and retreat. The effective basal melting anomaly therefore varies between the simulations (see Fig. 7d). These results highlight the need for further modeling studies and

observations on basal melting patterns near the grounding line.

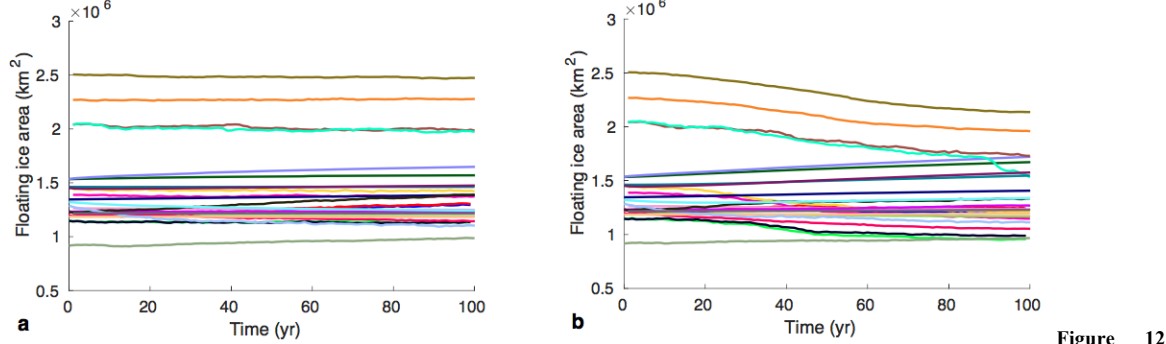

**Figure 12: Evolution of Antarctic ice shelf extent for the (a) *ctrl* and (b) *abmb* experiments.**

One objective of ISMIP6 and initMIP-Antarctica is to gather a large and diverse ice sheet modeling community. To facilitate participation of a large number of models, only two constraints were imposed: (1) the inclusion of both grounded and floating ice

and (2) the simulation of dynamic grounding line migration. This lack of constraints complicates the analysis of the simulation differences, since model parameters, input forcing, initialization techniques, and physical processes vary widely among models. Initialization methods that are based on the assimilation of present-day conditions usually have lower RMSE in the initial ice thickness and velocity compared to observations (Fig. 3) but larger trends in the *ctrl* experiment (Fig. 4a), while the opposite is true for models relying on paleoclimate spin-up or steady-state solution. This is similar to what was previously observed by

Nowicki et al. (2013a,b) and Goelzer et al. (2018). As the two approaches are complementary, models are starting to combine them by either following data assimilation with short relaxation periods, or by assimilating surface elevation during transient initialization to have an initial geometry more consistent with observations (Pollard and DeConto, 2012a). Combining the best of both approaches is an active field of research. Assimilating observations over longer time periods looks like a promising option, despite the technical challenges (Larour et al., 2014; Goldberg et al., 2015).

Representation of ice shelves and their connection to glaciers upstream is an outstanding cause of differences among models. Ice shelves are directly affected by changes in oceanic (Jacobs et al., 2011; Pritchard et al., 2012; Greenbaum et al., 2015; Wouters et al., 2015) and atmospheric (Scambos et al., 2000; Banwell et al., 2013; Munneke et al., 2014; Bassis and Ma,



2015) conditions, which impacts grounding line and ice front evolution (Favier et al., 2014; Joughin et al., 2014; Rignot et al., 2014; Bassis and Ma, 2015; Scheuchl et al., 2016; Christie et al., 2016; Seroussi et al., 2017). Ice shelf evolution over the past few decades has been complex, with large spatial and temporal variability (Depoorter et al., 2013; Rignot et al., 2013; Paolo et al., 2015; Christie et al., 2018) that is not fully understood and typically is not included in numerical models. Representation of

ice shelves varies significantly among models, resulting in large deviations in ice velocity, thickness, and applied basal melting applied. Significant progress was made over the past decade (Pattyn et al., 2017), but continued improvement of ice shelf representation in continental-scale models should remain a research priority.

The results presented in this study rely on simple atmospheric and oceanic forcings that are only loosely  based on RCP scenarios. Furthermore, many participating models did not use their full capabilities. To reduce model differences, for example

participants were asked to turn off surface-elevation feedback schemes, bedrock adjustment capabilities, and ice cliff failure. As a result, the iniMIP-Antarctica simulations are not projections of Antarctic evolution over the coming century and should not be compared with previous Antarctic simulations aiming to simulate this evolution (e.g., Ritz et al., 2015; Golledge et al., 2015). The next step of ISMIP6 will be assessment of Antarctic evolution under different scenarios forced with oceanic and atmospheric conditions derived from CMIP climate models;  experiments are now being designed. The initMIP-Antarctic simulations do,

however, illustrate the spread in ice sheet evolution (hence sea level) that is due to ice sheet model initial state and modeling choices (e.g., grounding line numerics, calving laws), and provide insight into uncertainty in simulations of sea level change.

## 6 Conclusions

The initMIP-Antarctica experiment, part of the Ice Sheet Model Intercomparison Project for CMIP6 (ISMIP6), had broad participation, with 25 model simulations submitted from 16 groups. Results are improved compared to previous similar exercises

of continental scale modeling of the Antarctic ice sheet, with enhanced representation of present-day conditions and ice mass loss trend. A first experiment performed with a simple surface mass balance anomaly forcing produces relatively robust results across the models, while a second experiment with a simple perturbation in basal melting rate under the ice shelves creates very large discrepancies in the ice sheet response. Variations in the representation of ice shelves, ice shelf melting, and numerical treatment of grounding lines cause this significant spread of results between the simulations. Including accurate representations

of ice shelves in continental scale models should therefore remain an important research subject in the coming years. All the experiments performed as part of initMIP-Antarctica are based on simplified anomaly forcings. Future projections of the Antarctic ice sheet evolution under different climate scenarios are currently being designed and will be the subject of future ISMIP6 modeling experiments.





## Appendix A: Outautos and output format

initMIP-Antarctica participants are required to provide outputs variables according to the data request plan. Three types of 2D fields are reported by modeling groups at 5-year intervals: state variables, flux variables and constants. Also, scalar outputs (e.g., total ice mass, ice mass above floatation, surface mass balance, basal melt) are reported every simulation year. Table A1 provides the complete list of requested variables. In addition to model output results, a README file describing model characteristics and details of the initialization procedure was requested from modeling groups for each simulation.

| Variable name | Type | Standard name | Unit |
|---|---|---|---|
| Ice sheet thickness | ST | land_ice_thickness | m |
| Ice sheet surface elevation | ST | surface_altitude | m |
| Ice sheet base elevation | ST | base_altitude | m |
| Bedrock elevation | ST | bedrock_altitude | m |
| Geothermal heat flux | CST | upward_geothermal_heat_flux_at_ground_level | $\mathrm{W\ m^{-2}}$ |
| Surface mass balance flux | FL | land_ice_surface_specific_mass_balance_flux | $\mathrm{kg\ m^{-2}\ s^{-1}}$ |
| Basal mass balance flux | FL | land_ice_basal_specific_mass_balance_flux | $\mathrm{kg\ m^{-2}\ s^{-1}}$ |
| Ice thickness imbalance | FL | tendency_of_land_ice_thickness | $\mathrm{m\ s^{-1}}$ |
| Surface velocity in x direction | ST | land_ice_surface_x_velocity | $\mathrm{m\ s^{-1}}$ |
| Surface velocity in y direction | ST | land_ice_surface_y_velocity | $\mathrm{m\ s^{-1}}$ |
| Surface velocity in z direction | ST | land_ice_surface_upward_velocity | $\mathrm{m\ s^{-1}}$ |
| Basal velocity in x direction | ST | land_ice_basal_x_velocity | $\mathrm{m\ s^{-1}}$ |
| Basal velocity in y direction | ST | land_ice_basal_y_velocity | $\mathrm{m\ s^{-1}}$ |
| Basal velocity in z direction | ST | land_ice_basal_upward_velocity | $\mathrm{m\ s^{-1}}$ |
| Mean velocity in x direction | ST | land_ice_vertical_mean_x_velocity | $\mathrm{m\ s^{-1}}$ |
| Mean velocity in y direction | ST | land_ice_vertical_mean_y_velocity | $\mathrm{m\ s^{-1}}$ |
| Ice surface temperature | ST | temperature_at_ground_level_in_snow_or_firn | K |
| Ice basal temperature | ST | land_ice_basal_temperature | K |
| Magnitude of basal drag | ST | magnitude_of_land_ice_basal_drag | Pa |
| Land ice calving flux | FL | land_ice_specific_mass_flux_due_to_calving | $\mathrm{kg\ m^{-2}\ s^{-1}}$ |
| Grounding line flux | FL | land_ice_specific_mass_flux_due_at_grounding_line | $\mathrm{kg\ m^{-2}\ s^{-1}}$ |
| Land ice area fraction | ST | land_ice_area_fraction | 1 |
| Grounded ice sheet area fraction | ST | grounded_ice_sheet_area_fraction | 1 |
| Floating ice sheet area fraction | ST | floating_ice_sheet_area_fraction | 1 |
| Total ice sheet mass | ST | land_ice_mass | kg |
| Total ice sheet mass above floatation | ST | land_ice_mass_not_displacing_sea_water | kg |
| Area covered by grounded ice | ST | grounded_land_ice_area | $\mathrm{m^2}$ |
| Area covered by floating ice | ST | floating_ice_shelf_area | $\mathrm{m^2}$ |
| Total SMB flux | FL | tendency_of_land_ice_mass_due_to_surface_mass_balance | $\mathrm{kg\ s^{-1}}$ |
| Total BMB flux | FL | tendency_of_land_ice_mass_due_to_basal_mass_balance | $\mathrm{kg\ s^{-1}}$ |
| Total calving flux | FL | tendency_of_land_ice_mass_due_to_calving | $\mathrm{kg\ s^{-1}}$ |
| Total grounding line flux | FL | tendency_of_grounded_ice_mass | $\mathrm{kg\ s^{-1}}$ |

**Table A1: Data requests for initMIP-Antarctica. ST: State variable, FX: Flux variable, CST: Constant**



## Appendix B: Model initialization

Below are descriptions of the initialization procedure performed by the different groups.

### ARC_PISM

We use the Parallel Ice Sheet Model (PISM) version 0.7.1. PISM is a "hybrid" ice sheet / shelf model that combines shallow
approximations of the flow equations that compute gravitational flow and flow by horizontal stretching (Bueler and Brown,
2009). We perform two sets of experiments with different initialisation procedures. In the first set (PISM-1,2), the simulations
are initialised from the end of a 120,000 year spin-up using paleoclimate forcing, whereas in the second set (PISM-3,4), the
simulations are initialised from the end of a 100,000 year spin-up using a constant climate forcing. Both procedures result in a
present-day ice sheet configuration that is in a thermally and dynamically evolved state, with "present-day" sea-level equivalent
volume of 58.35 m and 56.38 m, respectively. The combined stress balance of PISM allows for a treatment of ice sheet flow that
is consistent across non-sliding grounded ice to rapidly-sliding grounded ice (ice streams) and floating ice (shelves). As with
most continental-scale ice sheet models, we use flow enhancement factors for the shallow-ice and shallow-shelf components of
the stress regime (3.5 and 0.5 respectively for PISM-1,2, and 2.8 and 0.5 respectively for PISM-3,4), which allow us to adjust
creep and sliding velocities using simple coefficients. By doing so we are able to optimize simulations such that modelled
behaviour is consistent with observed behaviour. The junction between grounded and floating ice is refined by a sub-grid scale
parameterization (Feldmann et al., 2014) that smooths the basal shear stress field and tracks an interpolated grounding-line
position through time. This allows for much more realistic grounding-line motion, even with relatively coarse spatial grids, such
as the 16 km grid used in our experiments. We run duplicate experiments with the sub-grid melt turned off (PISM-1,3) or on
(PISM-2,4) in order to quantify the effect of this scheme. Surface mass balance is calculated using a positive degree day model
that takes as inputs air temperature and precipitation from RACMO2.1 (Lenaerts et al., 2012). In previous simulations (e.g.,
Golledge et al., 2015) we have derived evolving melt beneath ice shelves from the thermodynamic three-equation model of
Hellmer and Olber (1989), in which the melt rate is primarily controlled by salinity and temperature gradients across the ice–
ocean interface. For the simplified experiments presented here, however, we set a spatially uniform melt rate as an initial
condition and allow our modelled ice sheet to evolve in response to this.

### AWI_PISM

The simulations are performed with PISM version 0.7.3. For the 220 ka long spin-up simulations with paleo climatic forcing
(PISM1Pal), time slice anomalies for the Last Interglacial (LIG) and the Last Glacial Maximum (LGM) from the Earth System
Model COSMOS (Pfeiffer and Lohmann, 2016; Zhang et al., 2014) are used in addition to datasets for present-day (PD)
Antarctic climate (RACMO 2.3, van Wessem et al. (2014); WOA09, Locarnini et al. (2010). Time dependent and spatially
variable climate anomaly fields are interpolated during the PISM run between LIG, LGM and PD climate time slices with a
glacial index method (Sutter et al., 2016), where the glacial index is derived from Dome C deuterium depletion (Jouzel et al.,
2007). For the surface mass balance PISM's positive degree-day (PDD) scheme is used. Relative sea level forcing (Waelbroeck
et al., 2002) and bed deformation (Bueler et al., 2007) are applied during the paleo spin-up. In addition to the paleo spin-up a 100
ka long equilibrium-type spin-up (PISM1Eq) with steady present-day climate (ocean and atmosphere) and sea level is carried out
with isostatic bed deformation. Instead of precipitation and 2m air-temperature (PISM1Pal), surface mass balance and skin
temperature from RACMO2.3 are directly applied without the PDD scheme. The initial geometry for both spin-ups is Bedmap2
(Fretwell et al., 2013) and the geothermal flux is from Shapiro and Ritzwoller (2004). Basal shelf melt rates are calculated via a
quadratic form of the melt rate formula in Beckmann and Goosse (2003) using the extrapolated 3D ocean temperatures at the



depth of the ice shelf base. PISM's sub-grid grounding line scheme for basal sliding (Feldmann et al., 2014) is used in all simulations.

### CPOM_BISICLES

CPOM_BISICLES_A_500m is a block structured adaptive mesh finite element model based on a vertically-integrated stress

balance model (Cornford et al., 2013, 2016) and the basal friction physics of Tsai et al. (2015). Here, we make use of the adaptive mesh to maintain a resolution of 8 km in the slow moving interior, 1 km in ice streams, and 500 m at the grounding line. The initial state is based on ice thickness and bedrock elevation from Bedmap2 (Fretwell et al., 2013), modified according to mass conservation close to the grounding line to avoid the large unphysical thickening rates that would otherwise occur, especially in the Amundsen Sea Embayment. Ice temperature is taken from Pattyn (2010), and is held constant in time over the

course of the simulations. Effective viscosity $\varphi(x,y)$ and effective drag coefficients $\beta^2(x,y)$ are estimated by minimizing the mismatch between modelled speed the observed speed of (Rignot et al., 2011b), following the methods described in Cornford et al. (2015) The background ocean melt rate $M\_0(x,y,t)$ is defined so that the thinning rate is zero across the ice shelf, and varies in time accordingly, so that when a melt rate anomaly $M\_a(x, y, t)$ is applied, the ice shelf thinning rate is $M\_a(x, y, t)$.

CPOM_BISICLES_B is similar to CPOM_BISICLES_A, but does not allow accumulation onto the lower surface of the ice

shelf, so that the ice sheet thins where $div(uh) > 0$ even with no anomaly.

### DMI_PISM

The used Parallel Ice Sheet Model (PISM, version 0.7) utilizes a hybrid system (Bueler and Brown, 2009) combining the Shallow Ice Approximation (SIA) and Shallow Shelf Approximation (SSA) on a polar stereographic grid of 16 km. Monthly atmospheric forcing is deduced from sub-daily ERA-Interim reanalysis products (Berrisford et al., 2011; Dee et al., 2011)

covering the period 1979-2012. Its 2 m-air temperature drives the ice surface temperature, while the total precipitation is considered as snow accumulation due to negligible surface melting in Antarctica. Starting from the contemporary ice sheet geometry, both ice internal enthalpy and temperature evolve for 150,000 years for a fixed ice geometry due to surface and geothermal heat fluxes. Afterward the model runs freely for 25,000 years, so that the models updates continuously grounded ice margins, grounding lines and calving fronts. The calving parametrization exploits three sub-schemes for grid points at the ice

shelf margins: the Eigen-calving parameterization (Levermann et al., 2012), which utilizes the stress field divergence with the proportionality constant of $5 \times 10^{17}$, the ice shelf margin with a thickness of less than 150 m calve, and ice shelves that extent into the depth ocean calve. Assuming a constant ocean temperature of -1.7℃ and melting factor ($F_{melt}$ = 0.001) sub-shelf melting follows equation (5) in (Martin et al., 2011) and occurs only for fully floating grid points, while the grounding line position is determined on a sub-grid space (Feldmann et al., 2014). The basal resistance is described as plastic till for which the yield stress

is given by a Mohr-Coulomb formula (Bueler and Brown, 2009; Schoof, 2006).

### DOE_MALI

MPAS-Albany Land Ice (MALI) (Hoffman et al., 2018) uses a three-dimensional, first-order "Blatter-Pattyn" momentum balance solver solved using finite element methods. Ice velocity is solved on a two-dimensional map plane triangulation extruded vertically to form tetrahedra. Mass and tracer transport occur on the Voronoi dual mesh using a mass-conserving finite volume

first-order upwinding scheme. Mesh resolution is 2 km along grounding lines and in all marine regions of West Antarctica and in marine regions of East Antarctica where present day ice thickness is less than 2500 m to ensure that the grounding line remains in the fine resolution region even under full retreat of West Antarctica and large parts of East Antarctica. Mesh resolution





coarsens to 20 km in the ice sheet interior and no greater than 6 km in the large ice shelves. The horizontal mesh has 1.6 million cells. The mesh uses 10 vertical layers that are finest near the bed (4% of total thickness) and coarsen towards the surface (23% of total thickness). Ice temperature is based on results from Van Liefferinge and Pattyn (2013) and held fixed in time. The model uses a linear basal friction law with spatially-varying basal friction coefficient. The basal friction of grounded ice and the

viscosity of floating ice are inferred to best match observed surface velocity (Rignot et al., 2011b) using an adjoint-based optimization method (Perego et al., 2014) and then kept constant in time. The grounding line position is determined using hydrostatic equilibrium, with sub-element parameterization of the friction. Sub-ice-shelf melt rates come from Rignot et al. (2013) and are extrapolated across the entire model domain to provide non-zero ice shelf melt rates after grounding line retreat. The surface mass balance is from RACMO2.1 1979-2010 mean (Lenaerts et al., 2012). Maps of surface and basal mass balance

forcing are kept constant with time. The ice front position is fixed at the extent of the present-day ice sheet. After initialization, the model is relaxed for 99 years, so that the geometry and grounding lines can adjust.

### IGE_Elmer-Ice

For the momentum equations, we solve the shelfy-stream approximation. Using the methodology presented in Fürst et al. (2015, 2016), we rely on inverse methods to initialize the model to present-day conditions. We use the present-day ice sheet topography

and assimilate observed horizontal surface velocities to tune the basal friction coefficient and ice viscosity. The cost function also includes the mismatch between flux divergence and surface and basal mass balance. The initial friction coefficient and viscosity fields are kept constant during the forward simulations. The model is relaxed with a constant forcing for 20 years after the initialisation. For the control experiment the surface mass balance comes from the regional atmospheric model MAR (C. Agosta, personal communication) and is an averaged smb between 1979 and 2015. The basal melt rate depends on the difference

between ocean temperature and ocean freezing point and is a parameterisation by sector based on (Pollard and DeConto, 2012a). The bedrock topography is taken from Bedmap2 (Fretwell et al., 2013), except that we include two pinning points in contact with the bottom surface of Thwaites ice shelf using the bathymetry of Millan et al. (2017). The mesh is fixed and the resolution has been adapted to equi-distribute the interpolation error of the observed velocities and thickness with an additional criterion based on grounding line proximity. The horizontal resolution ranges between 1 km and 50 km.

### ILTS_SICOPOLIS

We use SICOPOLIS version 3.3-dev with either shallow-ice dynamics (ILTS_SICOPOLIS1) or hybrid shallow-ice–shelfy-stream dynamics (ILTS_SICOPOLIS2, Bernales et al. (2017)) for grounded ice and shallow-shelf dynamics for floating ice. Ice thermodynamics is treated with the melting-CTS enthalpy method (ENTM) by Greve and Blatter (2016). The ice surface is assumed to be traction-free. Basal sliding under grounded ice is described by a Weertman-type sliding law with sub-melt sliding

in the form of Sato and Greve (2012). The model is initialized by a paleoclimatic spin-up over 140000 years, forced by Vostok δD converted to ΔT (Petit et al., 1999), in which the topography is nudged towards the present-day topography to enforce a good agreement. In the future climate simulations, the ice topography evolves freely. For the last 2000 years of the spin-up and the future climate simulations, a regular (structured) grid with 8 km resolution is used. In the vertical, we use terrain-following coordinates with 81 layers in the ice domain and 41 layers in the thermal lithosphere layer below. The present-day surface

temperature is parameterized (Fortuin and Oerlemans, 1990), the present-day precipitation is by (Arthern et al., 2006) and (Le Brocq et al., 2010), runoff is modelled by the positive-degree-day method with the parameters by (Sato and Greve, 2012), the bed topography is Bedmap2 (Fretwell et al., 2013), and the geothermal heat flux is by (Purucker, 2012). Present-day ice shelf




basal melting is parameterized as a function of both the depth of ice below mean sea level and ocean temperatures outside the ice shelf fronts at 500 metres depth, tuned differently for eight Antarctic sectors (Greve and Galton-Fenzi, 2017).

### IMAU_IMAUICE

The finite difference model (de Boer et al., 2014) uses a combination of SIA and SSA solutions, with velocities added over

grounded ice to model basal sliding (Bueler and Brown, 2009). The model grid at 32 km horizontal resolution covers the entire Antarctic ice sheet and surrounding ice shelves. The grounded ice margin is freely evolving, while the shelf extends to the grid margin and a calving front is not explicitly determined. We use the Schoof flux boundary condition (Schoof, 2007) at the grounding line with a heuristic rule following Pollard and DeConto (2012b). For the initMIP experiments, the sea level equation is not solved or coupled (de Boer et al., 2014). We run the thermodynamically coupled model with constant present-day

boundary conditions to determine a thermodynamic steady state. The model is first initialised for 100 kyr using the average 1979-2014 SMB and surface ice temperature from RACMO 2.3 (van Wessem et al., 2014). Bedrock elevation is fixed in time with data taken from the Bedmap2 dataset (Fretwell et al., 2013), and geothermal heat flux data are from (Shapiro and Ritzwoller, 2004). We then run for 30 kyr with constant ice temperature from the first run to get to a dynamic steady state, which is our initial condition.

### JPL_ISSM

Model setup, as follows, is after Schlegel et al., 2018. The model domain covers present-day Antarctic Ice Sheet, and its geometry is interpolated from the Bedmap2 dataset (Fretwell et al., 2013), with addition refinement in the Amundsen Sea sector, Recovery Ice Stream, and Totten Glacier, after Morlighem et al. (2011) and Rignot et al. (2014). The forward simulations rely on a 2D Shelfy-Stream Approximation (MacAyeal, 1989) for stress balance, with a mesh resolution varying between 1 km at the

domain boundary and within the shear margins, 50 km in the interior, and a resolution of 8 km or finer within the boundary of all initial ice shelves. To estimate land ice viscosity, we compute the ice temperature based on a thermal steady state with 15 vertical layers (Seroussi et al., 2013), using three dimensional higher-order (Blatter, 1995; Pattyn, 2003) stress balance equations, observations of surface velocities (Rignot et al., 2011b), and basal friction inferred from surface elevations (Morlighem et al., 2010). Thermal boundary conditions are geothermal heat flux from (Maule et al., 2005) and surface temperatures from Lenaerts

et al. (2012). Steady state ice temperatures are vertically averaged, used as inputs in the ice flow law, and held constant over time. To infer the unknown basal friction coefficient over grounded ice and the ice viscosity of the floating ice, we use data assimilation (MacAyeal, 1993; Morlighem et al., 2010), to reproduce observed surface velocities from Rignot et al. (2011b). Then, we run the model forward for 2 years, allow the grounding line position and ice geometry to relax (Seroussi et al., 2011; Gillet-Chaulet et al., 2012). The grounding line evolves assuming hydrostatic equilibrium and following a sub-element grid

scheme (SEP2 in Seroussi et al., 2014). The ice front remains fixed in time during all simulations performed, and we impose a minimum ice thickness of 1 m everywhere in the domain. The surface mass balance and the ice shelf basal melt rates used in the control experiment are respectively from the 1979-2010 mean of RACMO2.1 (Lenaerts et al., 2012) and from the 2004-2013 mean after Schodlok et al. (2016).

### LSCE_GRISLI

The GRISLI model is a three-dimensional thermo-mechanically coupled ice sheet model originating from the coupling of the inland ice model of Ritz (1992) and Ritz et al. (1997) and the ice shelf model of Rommelaere (1996), extended to the case of ice streams treated as dragging ice shelves (Ritz et al., 2001). In the version used here, over the whole domain, the velocity field



consists in the superposition of the shallow-ice approximation (SIA) velocities for ice flow due to vertical shearing and the shallow-shelf approximation (SSA) velocities, used as a sliding law (Bueler and Brown, 2009). For the initMIP-Antarctica experiments, we used the GRISLI version 2.0 (Quiquet et al., 2018) which includes the analytical formulation of Schoof (2007) to compute the flux at the grounding line. Basal drag is computed with a power-law basal friction (Weertman, 1957). For this

study, we use an iterative inversion method to infer a spatially variable basal drag coefficient that insures an ice thickness as close as possible to observations with a minimal model drift (Le clec'h et al., 2018). The basal drag is assumed to be constant for the forward experiments.

The model uses finite differences on a staggered Arakawa C-grid in the horizontal plane at 16 km resolution with 21 vertical levels. Atmospheric forcing, namely near-surface air temperature and surface mass balance, is taken from the 1979-2014

climatological annual mean computed by the MAR version 3.6.4 regional atmospheric model (Agosta et al., 2018, in review). Initial sub-shelf basal melting rates are the regionally-averaged basal melting rates that ensure a minimal ice shelf thickness Eulerian derivative in a forward experiment with constant climate and fixed grounding line position. The initial ice sheet geometry, bedrock and ice thickness, is taken from the Bedmap2 dataset (Fretwell et al., 2013) and the geothermal heat flux is from Shapiro and Ritzwoller (2004).

### NCAR_CISM

The Community Ice Sheet Model (CISM, Lipscomb et al., 2018) uses finite element methods to solve a depth-integrated higher-order approximation (Goldberg and Sergienko, 2011) over the entire Antarctic ice sheet. The model uses a structured rectangular grid with uniform horizontal resolution of 4 km and five vertical σ–coordinate levels. The ice sheet is initialized with present-day geometry and an idealized temperature profile, then spun up for 30,000 years using 1979-2016 climatological surface mass

balance and surface air temperature from RACMO2 (van Wessem et al., 2018; Lenaerts et al., 2012). During the spin-up, basal friction parameters (for grounded ice) and sub-shelf melt rates (for floating ice) are adjusted to nudge the ice surface elevation toward present-day observations. This method is a hybrid approach between assimilation and spin-up, similar to that described by Pollard and DeConto (2012a). The geothermal heat flux is taken from Le Brocq et al. (2010). The basal sliding is similar to that of Schoof (2005), combining power-law and Coulomb behavior. The grounding line location is determined using hydrostatic

equilibrium and sub-element parameterization (Gladstone et al., 2010; Leguy et al., 2014, 2018). Basal melt is applied in grid cells that satisfy a flotation condition based on cell thickness and bed elevation; this includes some but not all cells intersected by the grounding line. The calving front is initialized from present day observations and thereafter is allowed to retreat but not advance. See Lipscomb et al. (2018) for more information about the model.

### PIK_PISM

With the Parallel Ice Sheet Model (PISM, Winkelmann et al., 2011, www.pism-docs.org, version 9ae1674 from August 2nd, 2017), we performed a paleoclimatic spin-up and an equilibrium simulation on a regular rectangular grid with 16 km and 8 km horizontal resolution, respectively. The vertical resolution increases from 130 m at the top of the domain to 20 m at the (ice) base, with a domain height of 6000 m. PISM uses a hybrid of the Shallow-Ice Approximation (SIA) and the two- dimensional Shelfy-Stream Approximation of the stress balance (SSA, MacAyeal, 1989; Bueler and Brown, 2009) over the entire Antarctic

Ice Sheet. The grounding line position is determined using hydrostatic equilibrium, with sub-grid interpolation of the friction at the grounding line (Feldmann et al., 2014). The calving front position can freely evolve using the Eigencalving parameterization (Levermann et al., 2012). PISM is a thermomechanically-coupled (polythermal) model based on the Glen-Paterson-Budd-





Lliboutry-Duval flow law (Aschwanden et al., 2012). The three-dimensional enthalpy field can evolve freely for given boundary conditions.

The model is initialized from Bedmap2 geometry (Fretwell et al., 2013), with precipitation from RACMOv2.3 1986-2005 mean (van Wessem et al., 2014) remapped from 27 km resolution and a parameterized ice surface temperature using the positive-

degree-day scheme (PDD, Huybrechts and de Wolde, 1999, modified by Martin et al. (2011)) for PIK_PISM3PAL. In contrast, PIK_PISM4EQUI uses SMB and temperature directly from RACMOv2.3 without PDD . Geothermal heat flux is from Shapiro and Ritzwoller (2004). We use the Potsdam Ice-shelf Cavity model (PICO, Reese et al., 2018) to calculate basal melt rate patterns underneath the ice shelves. We use observed ocean temperature and salinity mean values over the period 1979-2013 (Schmidtko et al., 2014) to drive PICO. The Mohr-Coulomb criterion relates the yield stress by parameterizations of till material

properties to the effective pressure on the saturated till (Bueler and van Pelt, 2015). Till friction angle is a shear strength parameter for the till material property and is optimized iteratively in the grounded-ice region such that the mismatch of equilibrium and modern surface elevation is minimized. This is analogous to the friction-coefficient optimization in Pollard and DeConto (2012a).

**PSU_PSUICE**

The Penn State University 3-D ice sheet model (PSUICE3D) is described in Pollard and DeConto (2012b), with updates in Pollard et al. (2015). The dynamics use a hybrid combination of vertically averaged SIA and SSA scaling. Floating ice shelves and grounding-line migration are included, with sub-grid interpolation for grounding-line position. The Schoof (2007) boundary-layer formulation is imposed as a condition on ice velocity across the grounding line. The model includes standard equations for

the evolution of ice thickness, and internal ice temperatures with 10 unevenly spaced vertical layers. Bedrock deformation under the ice load is modeled as an elastic lithospheric plate above local isostatic relaxation (ELRA). Basal sliding follows a Weertman-type power law, occurring only where the bed is close to the melt point. Basal sliding coefficients are determined by an inverse method (Pollard and DeConto, 2012a), iteratively matching ice surface elevations to modern observations. Atmospheric temperatures and precipitation are obtained from the ALBMAP climatology (Le Brocq et al., 2010), with an

imposed sinusoidal cycle for monthly air temperatures, interpolated to the ice sheet grid for surface mass balance calculations. Oceanic melting at the base of ice shelves depends on the squared difference between nearby 400-m depth climatological ocean temperature (Levitus, 2012), and the melt point at the bottom of the ice. "Standard" calving of ice shelves is included. InitMIP experiments are run without recently proposed mechanisms of hydrofracturing by surface meltwater, and structural failure of large ice cliffs (Pollard et al., 2015; DeConto and Pollard, 2016). The model grid size is 16 km, and two types of initialization are

used: (i) spun up to modern equilibrium (for 60 kyrs) with constant invariant model climate forcing, and (ii) run from 40 ka to modern using paleo climate forcing, and the model state at the end of that run is used.

**UCIJPL_ISSM**

We rely on inverse modeling to initialize the model to present-day conditions, following Morlighem et al. (2013). The mesh horizontal resolution varies from 3 km along the coast (in the vicinity of grounding lines and in shear margins) to 30 km inland,

and is extruded vertically in 10 layers. We use a Higher-Order stress balance (Pattyn, 2003) and an Enthalpy based thermal model (Aschwanden et al., 2012; Seroussi et al., 2013). We first perform an inversion of ice shelf viscosity, and then an inversion of basal drag under floating ice assuming thermo-mechanical steady state. Our geometry is primarily based on Bedmap-2 (Fretwell et al., 2013), with local improvements based on mass conservation in the Amundsen sea embayment, along the coast of Wiles land and on Recovery ice stream (Morlighem et al., 2011; Millan et al., 2017). The thermal model is



constrained by surface temperatures from Comiso (2000) and geothermal heat flux from Shapiro and Ritzwoller (2004), both included in the SeaRISE dataset (Shapiro and Ritzwoller, 2004; Nowicki et al., 2013b). The surface mass balance used in the control experiment is from RACMO 2.3 (van Wessem et al., 2014).

**ULB_f.ETISh**

The f.ETISh (fast Elementary Thermomechanical Ice Sheet) model (Pattyn, 2017) version 1.3 is a vertically integrated hybrid finite-difference (SSA for basal sliding; SIA for grounded ice deformation) ice sheet/ice shelf model with vertically-integrated thermomechanical coupling. The transient englacial temperature field is calculated in a 3d fashion. The marine boundary is represented by a grounding-line flux condition according to (Schoof, 2007), coherent a power-law basal sliding (power-law coefficient of 2). Model initialization is based on an adapted iterative procedure based on Pollard and DeConto (2012a) to fit the
model as close as possible to present-day observed thickness and flow field (Pattyn, 2017). The model is forced by present- day surface mass balance and temperature (van Wessem et al., 2014), based on the output of the regional atmospheric climate model RACMO2 for the period 1979-2011. The PICO model (Reese et al., 2018) was employed to calculate sub-shelf melt rates, based on present-day observed ocean temperature and salinity (Schmidtko et al., 2014) on which the initMIP forcings for the different basins are added. The model is run on a regular grid of 16 km with time steps of 0.05 year.

**VUB_AISMPALEO**

The Antarctic ice sheet model from the Vrije Universiteit Brussel derives from the coarse-resolution version used mainly in simulations of the glacial cycles (Huybrechts, 1990, 2002). It considers thermomechanically coupled flow in both the ice sheet and the ice shelf, using the shallow ice approximation/ shallow ice shelf approximation coupled across a one grid cell wide
transition zone. Basal sliding is calculated using a Weertman relation inversely proportional to the height above buoyancy wherever the ice is at the pressure melting point. The horizontal resolution is 20 km and there are 31 layers in the vertical. The model is initialised with a freely evolving geometry until steady-state is reached. The precipitation pattern is based on the Giovinetto and Zwally (2000) compilation used in Huybrechts et al. (2000), updated with accumulation rates obtained from shallow ice cores during the EPICA pre-site surveys (Huybrechts et al., 2007). Surface melting is calculated over the entire
model domain with the PDD scheme, including meltwater retention by refreezing and capillary forces in the snowpack(Janssens and Huybrechts, 2000). The sub-shelf basal melt rate is parameterised as a function of local mid-depth (485–700 m) ocean-water temperature above the freezing point (Beckmann and Goosse, 2003). A distinction is made between protected ice shelves (Ross and Filchner-Ronne) with a low melt factor and all other ice shelves with a higher melt factor. Ocean temperatures are derived from the LOVECLIM climate model (Goelzer et al., 2016) and parameters are chosen to reproduce observed average melt rates
(Depoorter et al., 2013). Heat conduction is calculated in a slab bedrock of 4 km thick underneath the ice sheet. Isostatic compensation is based on an elastic lithosphere floating on a viscous asthenosphere (ELRA model) but is not allowed to evolve further in line with the initMIP-Antarctica experiments.





**Appendix C: Modeled initial conditions**

| Model name | ice covered extent (10^6 km^2) | ice shelves extent (10^6 km^2) | ice mass (10^7 Gt) | ice mass above floatation (10^7 Gt) | surface mass balance (Gt) | basal melt (Gt) |
|---|---|---|---|---|---|---|
| ARC_PISM1 | 13.696 | 1.2348 | 2.5289 | 2.4656 | 2686 | 0 |
| ARC_PISM2 | 13.696 | 1.2348 | 2.5289 | 2.4656 | 2686 | 0 |
| ARC_PISM3 | 13.579 | 1.1466 | 2.3302 | 2.2785 | 2493 | 50 |
| ARC_PISM4 | 13.579 | 1.1463 | 2.3302 | 2.2785 | 2493 | 49 |
| AWI_PISM1Eq | 14.112 | 1.3885 | 2.4482 | 2.0979 | 2672 | 1233 |
| AWI_PISM1Pal | 14.669 | 1.4364 | 2.5602 | 2.1544 | 3061 | 1581 |
| CPOM_BISICLES_A | 13.654 | 1.5338 | 2.4118 | 2.0734 | 2144 | 2141 |
| CPOM_BISICLES_B | 13.654 | 1.5338 | 2.4118 | 2.0734 | 2144 | 2141 |
| DMI_PISM0 | 14.270 | 2.0408 | 2.1068 | 1.7873 | 3427 | 152 |
| DMI_PISM1 | 14.270 | 2.0411 | 2.1068 | 1.7873 | 3427 | 451 |
| DOE_MALI | 13.595 | 1.4623 | 2.3794 | 2.0467 | 2415 | 562 |
| IGE_ELMER | 13.590 | 1.3456 | 2.3885 | 2.0523 | 2515 | 784 |
| ILTS_SICIPOLIS1 | 13.609 | 1.1942 | 2.4050 | 2.0781 | 2020 | 456 |
| ILTS_SICIPOLIS2 | 13.591 | 1.2643 | 2.4092 | 2.0854 | 2015 | 508 |
| IMAU_IMAUICE32 | 14.174 | 1.2318 | 2.3535 | 2.0573 | 2706 | 0 |
| JPL1_ISSM | 13.905 | 1.4522 | 2.4382 | 2.1074 | 2337 | 986 |
| LSCE_GRISLI | 13.956 | 1.1991 | 2.4504 | 2.1081 | 2602 | 1565 |
| NCAR_CISM | 13.500 | 1.1850 | 2.3640 | 2.0422 | 2469 | 1125 |
| PIK_PISM3PAL | 14.556 | 1.2273 | 2.3574 | 2.0069 | 3191 | 583 |
| PIK_PISM4EQUI | 14.230 | 0.9168 | 2.3993 | 2.0466 | 2795 | 304 |
| PSU_EQNOMEC | 15.043 | 2.2700 | 2.4962 | 1.8772 | 2639 | 1278 |
| PSU_GLNOMEC | 15.003 | 2.5063 | 2.4970 | 1.8888 | 2679 | 1417 |
| UCIJPL_ISSM | 13.784 | 1.3217 | 2.4289 | 2.1142 | 2519 | 683 |
| ULB_FETISH1 | 13.889 | 1.6328 | 2.3972 | 2.0612 | 2660 | 2468 |
| VUB_AISMPALEO | 14.241 | 1.2167 | 2.5007 | 2.1603 | 2435 | 278 |

**Table C1: Simulated Antarctic initial ice-covered extent, ice-shelf extent, ice mass, ice mass above flotation, total surface mass balance and total basal melt.**



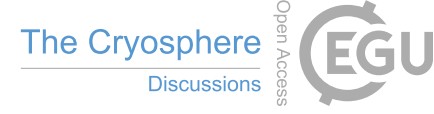

**Appendix D: Sea Level Contribution**

| Model name | *ctrl* | *asmb* | *abmb* |
|---|---|---|---|
| ARC_PISM1 | -112.9 | -287.1 | 66.5 |
| ARC_PISM2 | -115.0 | -300.6 | 154.7 |
| ARC_PISM3 | -5.5 | -151.3 | 88.6 |
| ARC_PISM4 | -2.4 | -154.9 | 215.4 |
| AWI_PISM1Eq | -22.1 | -172.3 | 22.7 |
| AWI_PISM1Pal | -48.4 | -213.0 | -4.5 |
| CPOM_BISICLES_A | 26.9 | -110.4 | 105.3 |
| CPOM_BISICLES_B | 83.3 | -54.3 | 169.1 |
| DMI_PISM0 | 0.8 | -140.1 | 94.4 |
| DMI_PISM1 | -4.1 | -140.0 | 108.8 |
| DOE_MALI | 167.3 | -26.6 | 249.9 |
| IGE_ELMER | -111.5 | -255.6 | -98.3 |
| ILTS_SICIPOLIS1 | -107.5 | -251.8 | -84.9 |
| ILTS_SICIPOLIS2 | -115.3 | -262.7 | -80.0 |
| IMAU_IMAUICE32 | 0.1 | -146.7 | 108.9 |
| JPL1_ISSM | -80.7 | -236.8 | 7.4 |
| LSCE_GRISLI | -167.6 | -324.6 | -149.6 |
| NCAR_CISM | 4.1 | -137.4 | 39.3 |
| PIK_PISM3PAL | -12.2 | -167.8 | 365.7 |
| PIK_PISM4EQUI | -19.8 | -181.4 | 407.0 |
| PSU_EQNOMEC | 12.7 | -112.0 | 47.5 |
| PSU_GLNOMEC | 16.2 | -108.9 | 50.7 |
| UCIJPL_ISSM | -243.6 | -400.0 | -178.5 |
| ULB_FETISH1 | -47.4 | -209.7 | -22.0 |
| VUB_AISMPALEO | -9.4 | -169.5 | 79.7 |

**Table D1: Antarctic contribution to sea level (mm sea level equivalent) at the end of the 100-year simulation for the three experiments and all submissions.**



***Acknowledgements.*** We acknowledge the Climate and Cryosphere (CliC) project and the World Climate Research Programme (WCRP) for their guidance, support, and sponsorship. We thank the CMIP6 panel members for their continuous leadership of the CMIP6 effort, and the Working Group on Coupled Modeling (WGCM) Infrastructure Panel (WIP) for overseeing the CMIP6 and ISMIP6 infrastructure and data request. Research was carried out at the Jet Propulsion Laboratory, California Institute of

Technology, under a contract with the National Aeronautics and Space Administration (80NM0018D0004). HS, NS and ER are supported by a grant from NASA Cryospheric Science Program. The National Center for Atmospheric Research is sponsored by the National Science Foundation. Ralf Greve was supported by Japan Society for the Promotion of Science (JSPS) KAKENHI grant numbers JP16H02224, JP17H06104 and JP17H06323. Christian Rodehacke (DMI) has received funding from the European Research Council under the European Community's Seventh Framework Programme (FP7) for research, Theme 6

Environment as part of the NACLIM (North Atlantic Climate) project (Grant Agreement 308299) as well as the Nordic Top-level Research Initiative (TRI) GREENICE (Impacts of Sea-Ice and Snow-Cover Changes on Climate, Green Growth and Society) funded by Nordforsk (Project no 61841). Support for MJH, SFP, and TZ was provided through the Scientific Discovery through Advanced Computing (SciDAC) program funded by the U.S. Department of Energy Office of Science, Biological and Environmental Research and Advanced Scientific Computing Research programs. The Ministry of Education, Culture and

Science (OCW), in the Netherlands, provided financial support for this study via the program of the Netherlands Earth System Science Centre (NESSC). The work of T.K. and A.H. has been conducted in the framework of the PalMod project (FKZ: 01LP1511B), supported by the German Federal Ministry of Education and Research (BMBF) as Research for Sustainability initiative (FONA). A.Q. acknowledges funding from the European Research Council grant ACCLIMATE no 339108. F.G.C. and J. B. (IGE) have received funding from the French National Research Agency (ANR) under the SUMER (Blanc SIMI 6) 2012

project ANR-12-BS06-0018. IGE-ELMER simulations were performed using HPC resources from GENCI-CINES (grant 2017-016066) and using the Froggy platform of the CIMENT infrastructure which is supported by the Rhone-Alpes region (grant CPER07_13 CIRA), the OSUG@2020 laBex (reference ANR10 LABX56), and the Equip@Meso project (reference ANR-10-EQPX-29-01). T.A. was supported by the Deutsche Forschungsgemeinschaft (DFG) in the framework of the priority program "Antarctic Research with comparative investigations in Arctic ice areas" by grant LE1448/6-1 and LE1448/7-1. Development of

PISM is supported by NASA grant NNX17AG65G and NSF grants PLR-1603799 and PLR-1644277. S.S. was supported by the FRS-FNRS MEDRISM project and the BELSPO MIMO project (Stereo III) . PH and JVB acknowledge support from the iceMOD project funded by the Research Foundation – Flanders (FWO-Vlaanderen).

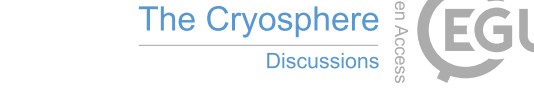

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
