# Peer review of "initMIP-Antarctica: An ice sheet model initialization experiment of ISMIP6"

_The Cryosphere, 2018_

## Referee Comment (RC1) · Anonymous Referee #1 · 17 Feb 2019

General Comments

This work contains the experimental setup for, and results from, the initMIP-Antarctica model inter-comparison project. 25 model simulations from 16 different participating groups are shown, representing a large amount of the cryospheric modelling community. Whilst the paper does not have a tremendous deal of new insights in and of itself, it does provide a fantastic overview of the current state of continent wide ice-sheet modelling. The paper is well written throughout providing a good summation of a large number of model results. The work is highly likely to be of great significance in the future.

A paper about a model inter-comparison project is going to be, by it's very nature, rather descriptive and as such I have rather few scientific criticisms. The one thing I

would like to see is for the authors to summarise and highlight somewhere what they think the key areas needed for model improvement in the future are. This is mentioned briefly in the conclusions, but I would like to see it expanded on.

Scientific comments

Pg 5, paragraph 2. Are there any specific requirements on how/whether to include ice calving/ ice front retreat/advance, or is that covered by not including ice cliff failure. Could do with clarification.

Pg 9, ln 28 . "Thickness and velocity variability is small, however, compared to the discrepancies between observations and models." Not sure I follow this. Are you trying to say that inter model variability is small compared to the absolute error between each model and observations? Fig. 3 would not back this up.

Pg 11, ln 8, Positive or negative thickness change? Does this include floating as well as grounded ice?

Technical errors

pg 5, ln 28, "participated to the" should be "paticipated in the"

pg 8, table 2 caption, start caption on new line

pg 8, ln 10, SS* is missing in the caption, is just (SS )

pg 12, ln 13. New line for start of section 4.3

pg 17, ln 36. Missing right bracket on ARC_PISM3

pg 18, ln 9, "shown on Fig 10" should be "shown by Fig 10"

pg 18, Fig 12. Fig 12 has no Fig number in caption.

---

## Referee Comment (RC2) · Jesse Johnson (Referee) · 1 Mar 2019

This paper aggregates some 25 modeling results from 16 different groups in order to identify the variability in model results with respect to surface mass balance and sub-shelf melting in Antarctica. The results are important because they demonstrate that while models have similar results for surface mass balance anomalies, there is considerable variability for anomalies in sub-shelf melting. The important differences are likely due to differences in both how the models are initialized an how sub-shelf melting is parameterized in models. Hence, the paper reports on both the present state-of-the art in terms of modeling, and it offers excellent suggestions for where modeling should go in the future if differences in modeling results are to be well understood.

[Figure]

The contribution is novel in that, unlike previous "SeaRISE" efforts, 1) the forcing data sets are now consistent with AR5 rather than AR4 climate forecasts, and 2) a larger number of models are participating and the models have a greater sophistication in terms of how they treat the physical mechanisms responsible for grounding line retreat and floating ice.

Given the importance and novelty of the results, I encourage rapid publication of the manuscript. It is well written and the complex results are presented in an accessible way. While reading the manuscript, I did have a few ideas that might make results easier for readers to reason about. I know that assembling so many results is a massive undertaking, and that most important finding are already easy to digest, so I leave it to the authors to decide if my suggestions are worth pursuing.

* page 1 line 35-36, even as a modeler I am not sure this is true. Let's not rule out semi-empirical approaches just yet. * page 3, line 0-10 - this stages the problems very well. * page 3, line 14 - I'm not sure you meet this objective. I'm not sure how high the bar is for 'enhance', but I finished the paper with plenty of questions as to what is responsible for the spread in results. Consider softening expectations? * page 4, lines 8-15 Unlike the previous paragraph, which I finished with a good understanding of the basis for SMB anomalies, I finished this paragraph unclear about what the anomalies in sub-shelf melt were based on. You take the present day melt rates estimated in Rignot 2013 and Deporter 2013 and double them? Ok, but how are the two references reconciled? Average? Consider rephrasing the contents of this paragraph. * page 8 - this table is at the center of a lot of what and how things that are to come are interpreted. Could the model names use the same color schemes are the figures to come. Also, DMI_PISM and ILTS_SICIPOLIS appear to be identical, at least according to the table. Could difference be noted here for clarity? * page 9, figure 2 - minor, but this is a continuously varying color map being used to represent 25 different things. Maybe it would be more clear if there were 25 discrete colors? * page 10, figure 3 - this is my most significant suggestion. It would be super helpful if the display of information

were clustered by a possible explanatory variable. For example, here, I think that the lower RMSEs in thickness and velocity are due to a assimilation as opposed to a spin up procedure, but I'm too lazy to compare models to the table. If you had the results boxed off according to initialization procedure, it would invite readers to do more speculation about the causes of differences. As it is, one just sees that some models are different from others, without the ability to reflect on cause. This criticism applies of much of what is to come in terms of 'clustering' model results according to something; initialization procedure, sub-grid parameterization, interpolation, etc. * page 10, figure 3 - I'm not sure I get much out of log *speed* (not velocity) as opposed to speed. * page 11, figure 4 - mention in caption that negative is growing the ice sheet? * page 14, figure 9 - I like this figure quite a bit. Again, clustering would help. * page 16, figure 11 - would it be helpful to place this along side figure 9? It's an interesting shift in sensitivity. * page 18, I really enjoyed the discussion, some strong points are made. However, I worry readers won't get this far. Consider a non-standard format of placing the discussion *before* the results? Probably a terrible idea, but the results do pacify the reader's attention.

Nice work pulling it all together.

---

## Author Comment (AC1) · 18 Mar 2019

Dear Dr. Matsuoka,

We are thankful to the two reviewers, who have provided helpful and insightful comments to our manuscript. All of their remarks have been considered; the response to reviewer details our responses to the reviewers and how we included their remarks.

The new version of the manuscript mostly includes additional clarifications of the text and small changes into a couple of figures and tables so that the models and their characteristics can be more easily visualized.

Best regards,

[Figure]

Helene Seroussi, on behalf of authors

Please also note the supplement to this comment:
https://www.the-cryosphere-discuss.net/tc-2018-271/tc-2018-271-AC1-
supplement.pdf

---

## Author Response (AR1)

**1 Reviewer #1**

General Comments

This work contains the experimental setup for, and results from, the initMIP-Antarctica model inter-comparison project. 25 model simulations from 16 different participating groups are shown, representing a large amount of the cryospheric modelling community. Whilst the paper does not have a tremendous deal of new insights in and of itself, it does provide a fantastic overview of the current state of continent wide ice-sheet modelling. The paper is well written throughout providing a good summation of a large number of model results. The work is highly likely to be of great significance in the future. A paper about a model inter-comparison project is going to be, by it's very nature, rather descriptive and as such I have rather few scientific criticisms. The one thing I would like to see is for the authors to summarise and highlight somewhere what they think the key areas needed for model improvement in the future are. This is mentioned briefly in the conclusions, but I would like to see it expanded on.

We thank the reviewer for his/her careful review. We agree that it is a good idea to better emphasize the critical areas that need improvements in the future so we now detail this more at in the discussion.

Scientific comments

Pg 5, paragraph 2. Are there any specific requirements on how/whether to include ice calving/ice front retreat/advance, or is that covered by not including ice cliff failure. Could do with clarification.

There is no ice front evolution requirement in the initMIP experiments. We added this to the text on page 5.

Pg 9, ln 28 . "Thickness and velocity variability is small, however, compared to the discrepancies between observations and models." Not sure I follow this. Are you trying to say that inter model variability is small compared to the absolute error between each model and observations? Fig. 3 would not back this up.

We were comparing the interannual variability of observed thickness and velocity that is small compared to the discrepancies between models and observations, so the exact initial model year probably does not matter. This sentence was rephrased.

Pg 11, ln 8, Positive or negative thickness change? Does this include floating as well as grounded ice?

This number is the ocean-induced basal melt, and only includes floating ice. We use positive numbers as we report the melt.

Technical errors

pg 5, ln 28, "participated to the" should be "paticipated in the" pg 8, table 2 caption, start caption on new line

Done

pg 8, table 2 caption, start caption on new line

Done

pg 8, ln 10, SS* is missing in the caption, is just (SS )

Done (the symbol used was wrong)

pg 12, ln 13. New line for start of section 4.3

Done

pg 17, ln 36. Missing right bracket on ARC_PISM3

Done

pg 18, ln 9, "shown on Fig 10" should be "shown by Fig 10"

Done

pg 18, Fig 12. Fig 12 has no Fig number in caption.

Done

**2   Reviewer #2 (J. Johnson)**

This paper aggregates some 25 modeling results from 16 different groups in order to identify the variability in model results with respect to surface mass balance and sub-shelf melting in Antarctica. The results are important because they demonstrate that while models have similar results for surface mass balance anomalies, there is considerable variability for anomalies in sub-shelf melting. The important differences are likely due to differences in both how the models are initialized an how sub-shelf melting is parameterized in models. Hence, the paper reports on both the present state-of-the art in terms of modeling, and it offers excellent suggestions for where modeling should go in the future if differences in modeling results are to be well understood.

The contribution is novel in that, unlike previous "SeaRISE" efforts, 1) the forcing data sets are now consistent with AR5 rather than AR4 climate forecasts, and 2) a larger number of models are participating and the models have a greater sophistication in terms of how they treat the physical mechanisms responsible

for grounding line retreat and floating ice. Given the importance and novelty of the results, I encourage rapid publication of the manuscript. It is well written and the complex results are presented in an accessible way. While reading the manuscript, I did have a few ideas that might make results easier for readers to reason about. I know that assembling so many results is a massive undertaking, and that most important finding are already easy to digest, so I leave it to the authors to decide if my suggestions are worth pursuing.

We thank Jesse Johnson for his careful review and his suggestions to improve the manuscript.

*page 1 line 35-36, even as a modeler I am not sure this is true. Let's not rule out semi-empirical approaches just yet.

We rephrased this sentence.

*page 3, line 0-10 - this stages the problems very well.

Thank you

*page 3, line 14 - I'm not sure you meet this objective. I'm not sure how high the bar is for 'enhance', but I finished the paper with plenty of questions as to what is responsible for the spread in results. Consider softening expectations?

We soften our expectations: this sentence is mainly here to explain that this is not a projection paper but rather a paper to demonstrate the impact of initial conditions and model choices on simulations.

*page 4, lines 8-15 Unlike the previous paragraph, which I finished with a good understanding of the basis for SMB anomalies, I finished this paragraph unclear about what the anomalies in sub-shelf melt were based on. You take the present day melt rates estimated in Rignot 2013 and Deporter 2013 and double them? Ok, but how are the two references reconciled? Average? Consider rephrasing the contents of this paragraph.

Yes, the two different datasets used to prepare the anomalies in basal melt were averaged. We also averaged the anomalies over each basin so that there is a spatially uniform anomaly applied under the floating ice per basin. This was clarified.

*page 8 - this table is at the center of a lot of what and how things that are to come are interpreted. Could the model names use the same color schemes are the figures to come. Also, DMI_PISM and ILTS_SICIPOLIS appear to be identical, at least according to the table. Could difference be noted here for clarity?

Adding colors similar to the other figures is a good idea so this was changed. The two ILTS_SICOPOLIS models differ by the stress balance approximation used while the two DMI_PISM models differ by the XXX (waiting for Christian). This was fixed in the table.

* page 9, figure 2 - minor, but this is a continuously varying color map being used to represent 25 different things. Maybe it would be more clear if there were 25 discrete colors?

It actually is a colormap with 25 discrete colors, but the conversion of the pdf into a jpeg blurred the transitions between the colors, so we will make sure to get a version of the figure with a better resolution in the final paper.

* page 10, figure 3 - this is my most significant suggestion. It would be super helpful if the display of information were clustered by a possible explanatory variable. For example, here, I think that the lower RMSEs in thickness and velocity are due to a assimilation as opposed to a spin up procedure, but I'm too lazy to compare models to the table. If you had the results boxed off according to initialization procedure, it would invite readers to do more speculation about the causes of differences. As it is, one just sees that some models are different from others, without the ability to reflect on cause. This criticism applies of much of what is to come in terms of 'clustering' model results according to something; initialization procedure, sub-grid parameterization, interpolation, etc.

The factor that has the most impact on the ability of models to reproduce observations is the inclusion of target values (either by data assimilation or target value at the end of spin-ups or steady state) during the initialization process. We marked the simulations that employ such target values with a ⋆ on Figure 3 to highlight the importance of this process.

* page 10, figure 3 - I'm not sure I get much out of log *speed* (not velocity) as opposed to speed.

Panel c shows the difference the log of the modeled speed and the log of the observed speed, so that fast flowing regions do not have an overwhelming weight in the error.

* page 11, figure 4 - mention in caption that negative is growing the ice sheet?

Done

* page 14, figure 9 - I like this figure quite a bit. Again, clustering would help.

It is important for us to be able to track individual models so that one can assess the individual performance of each simulation, even if it's difficult

and requires quite some time. Therefore, it is difficult to cluster simulations according to the initialization method or other model characteristic and we did not change the figure.

* page 16, figure 11 - would it be helpful to place this along side figure 9? It's an interesting shift in sensitivity.

This is a good point, we merged Fig.9 and Fig.11 into Fig.9 a and to facilitate the comparison between the two experiments.

* page 18, I really enjoyed the discussion, some strong points are made. However, I worry readers won't get this far. Consider a non-standard format of placing the discussion *before* the results? Probably a terrible idea, but the results do pacify the reader's attention.

It is indeed a rather long paper and we considered moving the discussion before the results. However, many conclusions made in the discussion refer to points explained in the results section, which would make the paper even longer if we had to explain things twice. We expect readers from different parts of the community to be interested in different aspects of this manuscript and therefore to focus on one or the other section.

Nice work pulling it all together.

[revised manuscript text omitted]